# Current Approaches for the Prevention and Treatment of Acute and Chronic GVHD

**DOI:** 10.3390/cells13181524

**Published:** 2024-09-11

**Authors:** Attilio Olivieri, Giorgia Mancini

**Affiliations:** 1Clinica di Ematologia, Università Politecnica delle Marche Ancona, 60126 Ancona, Italy; 2Department of Hematology, AOU delle Marche Ancona, 60126 Ancona, Italy; giorgia.mancini@ospedaliriuniti.marche.it

**Keywords:** transplantation, graft versus host prophylaxis, graft versus host treatment

## Abstract

Whereas aGVHD has strong inflammatory components, cGVHD displays autoimmune and fibrotic features; incidence and risk factors are similar but not identical; indeed, the aGVHD is the main risk factor for cGVHD. Calcineurin Inhibitors (CNI) with either Methotrexate (MTX) or Mycophenolate (MMF) still represent the standard prophylaxis in HLA-matched allogeneic stem cell transplantation (HSCT); other strategies focused on ATG, Post-Transplant Cyclophosphamide (PTCy), Abatacept and graft manipulation. Despite the high rate, first-line treatment for aGVHD is represented by corticosteroids, and Ruxolitinib is the standard second-line therapy; investigational approaches include Microbiota transplant and the infusion of Mesenchymal stem cells. GVHD is a pleiotropic disease involving any anatomical district; also, Ruxolitinib represents the standard for steroid-refractory cGVHD in this setting. It is a pleiotropic disease involving any anatomical district; also, Ruxolitinib represents the standard for steroid-refractory cGVHD in this setting. Extracorporeal Photopheresis (ECP) is still an option used for steroid refractoriness or to achieve a steroid-sparing. For Ruxolitinib-refractory cGVHD, Belumosudil and Axatilimab represent the most promising agents. Bronchiolitis obliterans syndrome (BOS) still represents a challenge; among the compounds targeting non-immune effectors, Alvelestat, a Neutrophil elastase inhibitor, seems promising in BOS. Finally, in both aGVHD and cGVHD, the association of biological markers with specific disease manifestations could help refine risk stratification and the availability of reliable biomarkers for specific treatments.

## 1. Introduction

Graft-versus-host disease (GVHD) is still the leading cause of non-relapse mortality occurring after allogeneic hematopoietic stem cell transplantation (HSCT) [1,2].

Since 1982, acute (aGVHD) and chronic GVHD (cGVHD) have been described as two distinct diseases in the dog model: whereas aGVHD has strong inflammatory components, cGVHD displays autoimmune and fibrotic features; however, this paradigm is not absolute.

Clinically. It is often challenging to separate aGVHD from cGVHD as these two entities’ temporal development and clinical manifestations can also overlap. Initially, the GVHD clinical manifestations have been chronologically categorized into acute (early onset, within 3 months after HSCT) and chronic (with onset later than 3 months after HSCT); the overlap syndrome has been introduced in order to cover the clinical pictures of cGVHD patients with associated manifestations of aGVHD [3,4].

Typically, acute GVHD may involve three main districts, represented by skin, gastrointestinal tract (GIT) and liver, but other targets of aGVHD have been identified, such as bone marrow, thymus, lung, ovary, central nervous system (CNS) and endothelium [5]. Skin is the most commonly involved GVHD target organ with the typical morbilliform rash; the other skin changes may include petechiae and hyperpigmentation, fluid-filled bullae blisters or small dry skin patches with desquamation. aGVHD involvement of the upper GIT tract is frequent but variable, with incidence ranging from 24–60%. Upper GIT aGVHD is characterized by anorexia, nausea, vomiting and dyspepsia; this form often resolves spontaneously or quickly responds to steroids; on the contrary, lower GIT GVHD is the kind of involvement associated with higher non-relapse mortality; it is typically characterized by choleriform diarrhea and/or hematochezia, severe abdominal pain or ileus. The liver is the least frequent manifestation; increased serum bilirubin levels are the unique hallmark, although not specific, as several causes of hyperbilirubinemia should be excluded in these patients; if hyperbilirubinemia develops contemporaneously or after an aGVHD manifestation in the skin or GIT, a liver GVHD should be considered in absence of an identified alternative cause; transaminitis without hyperbilirubinemia is not a specific liver GVHD hallmark unless he presence of GVHD is confirmed by liver biopsy [2].

cGvHD is a pleiotropic disease, showing clinical features reminiscent of autoimmune disorders and potentially involving each organ, including diagnostic, distinctive and common manifestations (these latter are shared with aGVHD). Its presentation may be progressive without separation from aGVHD after the resolution of aGVHD or de novo. cGVHD often shows scleroderma-like manifestations of skin and fasciae and visceral involvement (typically lung and esophagus); the other most frequently involved anatomical districts are represented by oral mucosa, eye, GIT, genital tract and joints. cGVHD affects the long-term outcomes of transplanted patients by increasing morbidity and mortality and is associated with a reduced quality of life. It often requires long-term immunosuppressive therapy, causing severe side effects and toxicities [6]. Today, the timing of the GVHD onset is no longer considered a distinctive feature: while in the past, any manifestation that occurred within day 100 was considered as aGVHD, after day 100, it was considered as cGVHD. After the NIH 2005 consensus [3], acute and cGVHD are currently identified according to the kind of clinical manifestations, regardless of onset time. A joint EBMT-CIMBTR-NIH task force [7] has standardized the terminology of diagnosis, staging and treatment response for different subtypes of GVHD. While these recommendations clearly outline the diagnosis of acute (either classical or “late acute”) and chronic GVHD (both classical and “early chronic” subtypes), some degree of uncertainty still concerns the so-called “overlap syndromes”, i.e., simultaneously showing features of acute and cGVHD (Table 1). The category defined as “overlap chronic GVHD”, which has been recently introduced [4], includes a subtype of cGvHD often associated with a poor prognosis; in this setting, we can typically observe patients with cGVHD manifestations, accompanied by acute GIT manifestations (anorexia, nausea, vomiting, diarrhea); however, skin manifestations of aGvHD (maculo-papular-erythematous rash) can be difficult to differentiate from those of cGvHD. Similarly, hyperbilirubinemia, which suggests liver involvement, cannot be unequivocally attributable to either an acute or a chronic process [4,8].

In this review, mainly dedicated to Transplant physicians, we outline the standard recently achieved and the main progress in the prevention and treatment of acute and cGVHD, addressing the most significant recent experimental evidence leading to a possible change in the current clinical practice, focusing the emerging clinical and translational data, focusing the most relevant unmet needs and the relevant pathogenetic pathways that could be targeted with new compounds.

Risk factors of acute and cGVHD

The incidence and risk factors of acute and cGVHD are similar (Table 2) but not identical, with aGVHD being the main risk factor for cGVHD development [9,10]. During the past three decades, several studies have identified risk factors associated with acute and cGVHD (Table 2).

In a Japanese retrospective study, which included 4818 adult patients receiving HSCT, the incidence of cGVHD at 2 years was 37%. The following factors were associated with the development of cGVHD: i—female donor/male recipient; ii—CMV-antibody seropositivity; iii—matched related peripheral blood grafts vs. matched related BM grafts; iv—no in vivo T-cell depletion and the occurrence of grade II–IV aGVHD. Among these factors, the association with aGVHD occurrence was consistently significant across donor subtypes. The use of the Cord Blood source was associated with a low incidence of extensive cGVHD [14].

In a large prospective study, 775 patients receiving HSCT were evaluated according to the NIH consensus criteria, and the cumulative incidence of aGVHD was 44.7%, consisting of classic aGVHD (n = 320) and late-onset (n = 26). Multivariate analyses revealed that younger age, unrelated donors and acute leukemia diagnosis were significant risk factors for aGVHD [20].

Myeloablative and total body irradiation-based regimens, associated with more significant tissue damage, have also been linked to higher rates of both acute and cGVHD [11,15,16,17,18,19].

The risk factors for aGVHD and cGVHD according to the NIH consensus criteria were compared in 2941 recipients of HSCT, and this analysis identified other common risk factors such as donor-age, severe infections during the peri-transplant period and previous administration of Donor Lymphocyte Infusion (DLI) [10].

In 2012, a randomized clinical trial (RCT) comparing bone marrow (BM) source vs. peripheral blood stem cell source (PBSC) in MUD recipients showed that PBSC recipients had higher rates of cGVHD and were more likely to require immunosuppressive therapy at 2 years, while there was no difference for aGVHD incidence [12]. In a more recent retrospective study [13], rates of grades II–IV aGVHD (58% versus 45%) and cGVHD (56% versus 42%) were significantly higher with PBSC than with BM transplants. A metanalysis showed that both aGVHD and cGVHD were significantly increased when comparing PBSC and BM transplants [21].

As for the setting of haploidentical HSCT, a matched pair analysis conducted by O’Donnell in the non-myeloablative setting, no differences in terms of aGVHD, cGVHD, or Overall Survival (OS) have been found between BM and PBSC source [22].

In a more recent study, Gooptu [23] compared the use of post-transplant cyclophosphamide (PTCy) in haploidentical and MUD (matched unrelated donor) settings: the use of PBSC was associated with an increased risk of both aGVHD (2–4) and cGVHD in the reduced intensity conditioning (RIC) setting, and of cGVHD risk but only in the myeloablative conditioning (MAC) regimen, as reported by others [24,25]. A meta-analysis, including 13 studies regarding the PBSC versus BM analysis and 11 for the MAC versus RIC analysis (for 5965 patients), showed that both aGVHD and cGVHD were significantly increased with PBSC [11]. No differences were observed in terms of OS and non-relapse mortality (NRM), while, regardless of the intensity of the conditioning regimen, the use of PBSC, compared to BM, was associated with a higher risk of aGVHD II–IV and severe cGVHD [26].

Nevertheless, the graft source in the setting of haploidentical transplantation using GVHD prophylaxis with PTCy still remains an area of debate [27,28].

## 2. Pathophysiology of Acute and Chronic GVHD

aGVHD is a process initiated by innate immune cells, triggered during the cytokine storm induced by tissue injury, while in the late phase, the aGVHD development is amplified by adaptive immune responses. On the other hand, cGVHD is characterized by thymus damage with loss of central and peripheral tolerance, T helper type 2 responses, autoimmunity (mediated by aberrant B-cell expansion) and often fibrotic damage (mediated by macrophages, inducing fibroblast activation) with tissues remodeling.

The main immune perturbations in cGVHD are represented by donor alloreactive T-cell priming against host tissues, with the progressive development of uncontrolled aberrant B cells allo/autoreactivity in acute and chronic inflammation. Preclinical and clinical findings in human cGVHD documented a severe, combined cellular and humoral immune deficiency attributed to decreased CD4+ T reg cells and altered B cell function [11,29,30].

The aGVHD development recognizes a sequence of steps, starting from conditioning-induced tissue damage, with donor T-cell priming and expansion, involving the recruitment of neutrophils and macrophages, to the effector stage and tissue damage [11,31,32].

The first step involves innate immune cells, triggered during the cytokine storm and by tissue injury; the cytokine cascade is activated by PAMPS (pathogen-associated molecular patterns) and by DAMPS (damage-associated molecular patterns) release, with increased production of inflammatory cytokines such as IL12, IL23 and IL6; intestinal commensal bacteria and uric acid contribute to the inflammasome and this inflammatory network leads to increased IFNg production by innate lymphoid cells and by recipient T cells. The second step is characterized by donor T cell priming, facilitated by the upregulation of MHC antigens on host target cells, inducing a specific donor T cell alloreactivity against the host tissues. In vivo experiments demonstrated an early migration of allogeneic cells first to peripheral lymphoid organs and then to GVHD target organs.

In animal models, activated donor dendritic cells expand and migrate to mesenteric lymph nodes, promoting T cell priming and gut homing in a cascade driven by GM-CSF secretion. T-bet is the master transcription regulator of helper T lymphocytes (Th1) by increasing their responsiveness to IL-12 and inducing the effector T cell migration in the target organs. In the colon donor CD103(+) CD11b(–), dendritic cells (DCs) migrate to mesenteric lymph nodes, inducing specific homing signatures on donor T cells, initiating the GIT GVHD [33,34]. Microbiome-derived metabolites that modulate GIT damage, including butyrate, provide a connection between microbiome changes and the development of acute intestinal GVHD [35,36,37].

During the third step, the primed T cells become able to recognize and attack the target recipient tissues: skin, gut and biliary ducts; T helper 17 (Th17)/T cytotoxic 17 (Tc17) polarization is pivotal in this phase: Th17/Tc17 cell expansion is mediated by IL-6. Figure 1 illustrates this sequence, focusing on two additional steps, represented by the immune effector trafficking and homing (step 4), followed by the recruitment of non-immune effectors, such as monocytes and neutrophils, to inflammatory sites of the specific organ targets (step 5); Th17 polarized cells amplify aGVHD by generating inflammatory cytokines.

Similarly, in the cGVHD pathophysiology (Figure 2), three main distinct steps have been identified: (a) early inflammation due to tissue injury; (b) thymic damage with loss of tolerance and T/B cell dysregulation with autoimmune phenomena; (c) fibrosis with tissues remodeling.

Like aGVHD, the initial phase of cGVHD begins with damage to host tissues by conditioning and release of inflammatory cytokines. The damaged tissues release DAMPs, and the inflammatory cytokines stimulate donor alloreactive T cells, driving helper Th17/Tc17 polarization/expansion, amplifying the cytotoxic damage to several organs, including the endothelium; the IL-33 receptor ST2, a biomarker for acute and cGVHD, is released in response to endothelial damage [30,32]. The second step is characterized by thymic injury with activation of effector T cells, B cells, and Antigen-Presenting Cells (APC), generating a chronic inflammation auto-maintained by Th17 cells that have escaped immune regulation. One recurrent finding in cGVHD patients is the low naïve Treg frequency [38]; at the same time, the Th17/Tc17 cells emerge as the major driver of cGVHD, secreting pro-inflammatory cytokines, providing a cellular reservoir for effector alloimmune cells and supporting the T follicular helper (Tfh)-driven aberrant immune response [39].

cGVHD patients typically show altered B cell homeostasis with loss of regulatory B cells and aberrant B cell activity; the loss of central and peripheral tolerance leads to impaired Treg/regulatory T cells) and Breg cells (B regulatory cells); during this process Tfh and germinal centers B cells cooperate to generate long-lived, IL-6 driven, aberrant plasma cells and memory B cells able to produce auto/alloantibodies (Abs); elevated B-cell activating factor (BAFF) levels promote survival of autoreactive and alloreactive B cells [40,41]; BCR (B cell receptor) signaling is also required for B-cell differentiation and survival, promoting the expression of BAFF receptors [41]. Some auto-Abs, such as ANA, anti-double-stranded DNA, and activating antibodies, such as anti-PDGFR inducing tyrosine phosphorylation and fibrosis [42,43], have been associated with cGVHD as well as Abs anti-Y-chromosome encoded epitopes in male recipients of stem cell grafts from female donors [44].

The third step of cGVHD is characterized by fibrotic damage and tissue remodeling; fibrotic manifestations of cGVHD are often refractory to several treatments and, in some cases, are judged irreversible. Profibrotic cytokines such as TGF-beta and PDGF are upregulated and play key roles in fibrogenesis, stimulating the aberrant fibroblast activation with exaggerated collagen matrix production [11,32,33,42,45,46].

Macrophages are pivotal for the generation of fibrosis; a milieu rich in IL-17 and GM-CSF promotes adhesion and elicits a pro-inflammatory transcriptome in macrophages. The macrophage polarization is mediated by high levels of Fc-g receptors, which mediate the opsonization of antibody-coated targets, in turn generating TGF-beta, which stimulates collagen production; moreover, stimulating anti-PDGFR antibodies contribute to fibroblast activation [32,47].

## 3. GVHD Prevention

Despite the extensive use of prophylaxis regimens, aGVHD occurs in approximately 20–50% of transplanted patients [2] and is a significant cause of NRM after allogeneic HSCT; cGVHD occurs in 30% to 40% of patients who have undergone allogeneic hemopoietic stem cell transplantation [1,10] and deeply affects late NRM and the long-term quality of life [17]; therefore, the prevention of GVHD still represents an unmet medical need. One of the most critical issues in GVHD prevention is the selection of the most suitable donor: HSCT from HLA-identical matched sibling remains the gold standard, being associated with the lowest rates of acute and cGVHD. HLA mismatching at HLA-A, HLA-B, HLA-C, and HLA-DRB1 is associated with increased rates of GVHD and decreased survival [48]. Mismatches at HLA-DQB1, HDRB3/4/5 and others may increase aGVHD risk; non-permissive HLA-DPB1 mismatches should be avoided [49,50].

The other strategies to reduce acute and cGVHD incidence have focused on preparative regimens and graft manipulation.

According to the pathophysiological three-step sequence proposed by Ferrara [31], several checkpoints can be targeted for prophylaxis of aGVHD: alloreactive T-cell depletion/inactivation, anti-homing compounds or anti-cytokine therapy; recently, the alteration of microbiota emerged as a pivotal driver of the GIT GVHD [34,51].

As the alloreactive T-lymphocytes represent the main effectors of aGVHD, the backbone of prophylactic strategies is still based on alloreactive T-cell ex vivo or in vivo depletion/inactivation, being the anti-lymphocyte globulin (ATG) and PTCY the most popular tools; another target for GVHD prophylaxis is the T-cell receptor (TCR) signal which can be blocked by the CNI, which inhibits the proliferation and activation of T cells [52].

CNI (Tacrolimus and Cyclosporine) in combination with either Methotrexate (MTX) or Mycophenolate (MMF) still represent the main standard prophylaxis in HLA-matched HSCT; MMF depletes guanosine nucleotides in T and B lymphocytes and inhibits their proliferation, thereby suppressing cell-mediated immune responses and antibody formation. In two old RCTs, the combination of Tacrolimus (Tac)/MTX was significantly superior to Cyclosporine (CyA)/MTX in preventing grade II–IV aGVHD and extensive cGVHD both in HLA-matched sibling and in MUD transplants, but not in terms of OS [53,54]. A subsequent RCT comparing Tac/MTX versus Tac/MMF showed the superiority of Tac/MTX in preventing severe aGVHD, particularly in the MUD setting [55]. More recently, a Swedish group reported the results of an RCT in 215 patients receiving HSCT from matched donors with RIC, comparing Tac/MMF versus CSA + MMF: no significant difference in the cumulative incidence of grades II–IV aGVHD (41% versus 51%) or grades III–IV aGVHD (13% versus 7%) have been observed between the two groups; NRM, RFS (relapse-free survival) and OS were comparable [56]. A second RCT has been conducted by a Japanese group, comparing CyA and Tac in 107 patients with target blood concentrations of 500 and 15 ng/mL, respectively, to prevent aGVHD after MUD HSCT. The incidences of grade II–IV and grade III–IV aGVHD were 40% and 7.5% for the CyA group and 33 and 9% for the Tac group, respectively, without significant differences [57], suggesting a substantially similar efficacy between these two CNIs. In conclusion, in the context of traditional prophylaxis, according to the results of RCT, no significant advantages appear to emerge for one CNI over the other; at the same time, MMF is currently indicated for the combined prophylaxis with CNI and PTCY in the haploidentical setting or as an alternative to MTX in the MRD (matched related donor)/MUD setting, but preferentially in the context of reduced toxicity conditioning.

As the GVHD development is associated with lower levels of Tregs, Sirolimus emerged as a potential tool; Sirolimus is a mTOR inhibitor that inhibits effector T-lymphocytes and in in-vitro studies appeared to spare regulatory T-lymphocytes. It has a distinct toxicity profile compared to Tac and is not nephrotoxic. An RCT conducted in patients receiving myeloablative transplants from HLA-matched donors compared the combination of Sirolimus + Tac versus Tac/MTX. There was no difference in grades II–IV aGVHD and cGVHD, but better grade III–IV aGVHD outcomes with Sirolimus/Tac were seen, while NRM and OS were similar [58]. Recently, the addition of Sirolimus was also evaluated in the RIC setting in two RCTs [59,60]. A recent meta-analysis, with a total sample size of 1.673 cases, showed that Sirolimus containing prophylaxis could reduce grade II–IV aGVHD but was not associated with an improvement of grade III–IV aGVHD and OS; however, patients receiving Sirolimus-based regimens had increased Transplant-associated microangiopathy and VOD (veno-occlusive disease) incidence [61].

Rabbit ATG effectively depletes T-cells, preserving regulatory T-cells [62,63,64,65]. Two randomized studies compared Fresenius ATG prophylaxis versus standard prophylaxis in the MUD setting and the HLA-matched sibling setting with PBSC source, showing a significant reduction of cGVHD incidence [66,67]. Another study comparing Tac/MTX ± ATG in a myeloablative MUD setting showed a significant reduction in grade II–IV aGVHD and cGVHD with ATG; however, NRM and OS were reduced in the ATG arm [68]. Two more RCTs, conducted comparing the other rabbit ATG brand (Thymoglobuline), showed a significant reduction of cGVHD risk [69,70,71].

A recent systematic review evaluated seven RCTs: overall, the use of ATG significantly reduced aGVHD grade II–IV risk (RR 0.68); ATG significantly lowered cGvHD incidence (RR of 0.53) in eight studies, including 1273 patients. However, NRM did not seem to be affected by ATG [72].

Alemtuzumab is a humanized IgG monoclonal antibody targeting the CD52 antigen, which is present on the surface of peripheral blood cells, including both B and T cells, but sparing the hemopoietic progenitor cells [73]. In the UK, several groups incorporated Alemtuzumab (Campath-1H) as part of a Fludarabine-based protocol, and a low incidence of GVHD has been reported in both related and unrelated donor transplantation. This antibody has also been recommended in the GVHD prophylaxis in the EBMT/ESID inborn errors working party guidelines for hematopoietic stem cell transplantation for inborn errors of immunity [74].

Antibodies such as ATG and Alemtuzumab exert variable effects based on dosage, timing of administration, and formulation. Early studies employed high doses of Alemtuzumab, but recently, a safe de-escalation of the Alemtuzumab dose has been successfully proposed. Recently, an RCT compared two GVHD prophylaxis arms: high-dose Alemtuzumab/Cyclosporine (AC) and Tacrolimus/Methotrexate/Sirolimus (TMS): the incidence of severe cGVHD was significantly lower with AC versus TMS at 1- and 5-years (0% versus 10.3% and 4.5% versus 28.5%, respectively); however, despite the significant prevention of cGVHD seen with AC, the benefit was offset by increased relapse, as well as by the increased infection rate [75].

The advent of PTCy allowed a revolutionary change by expanding the practice of Haploidentical transplantation [76]. PTCy preferentially eliminates alloreactive donor T cells and induces peripheral tolerance through clonal deletion and Treg suppression; recent data suggest that alloreactive T cells are not eliminated but show reduced proliferation and impaired function [77,78]. In some retrospective studies in the haploidentical setting, severe acute and cGVHD seemed to be reduced after PTCy [25,79]. Based on its success in the haploidentical setting, the PTCy-based GVHD prophylaxis regimens have been tested beyond the haploidentical setting to replace conventional CNI-based regimens in the MRD and the MUD setting. A large retrospective CIBMTR registry study [23] comparing PTCy in 284 adults receiving MUD transplant versus PTCy in 2036 adults receiving haploidentical transplant with PTCy showed higher grade III–IV aGVHD in the haploidentical setting suggesting that MUD transplant is still the gold standard in this setting.

In a recent RCT conducted in 346 patients with acute leukemia and receiving HLA-matched donor transplant after myeloablative conditioning, three different regimens of GVHD prophylaxis have been compared: i—CD34-selected PBSC; ii—PTCy after BM graft; iii—Tac-MTX after BM graft (standard control). The CD34 selection prophylaxis was associated with lower moderate-severe cGVHD but higher NRM. PTCY was associated with comparable cGVHD and survival outcomes to control and a trend toward lower disease relapse [24]. The last RCT compared PTCy + Tac + MMF versus standard prophylaxis with Tac-MTX in 431 leukemia patients receiving reduced-intensity or non-myeloablative PBSC-HSCT from identical siblings or MUD. The cumulative incidence of grade III or IV aGVHD was lower in the PTCy group (6.3%) than in the standard-prophylaxis group (14.7%). Similarly, the cumulative incidence of cGVHD at 12 months was lower with PTCy (21.9% versus 35.1%) [80].

Other strategies to reduce acute and cGVHD have focused on graft manipulation, including the depletion of a/b T cells, the CD34+ positive selection or the selective depletion of CD45RA naïve T cells. Clinical data supported the benefit of CD34+ selection for the reduction of acute and cGVHD, albeit with high rates of infections for adults with hematological malignancies [81]. More recent studies included pediatric patients receiving haploidentical transplants with higher CD34+ doses (19 and 21.5 × 106/kg) and higher T cell doses (1.4 and 4.7 × 104); despite the high incidence of graft failure, low rates of aGVHD and cGVHD, have been reported [82,83].

Selective depletion of CD45RA naïve T cells preserves the CD34+ fraction, critical for engraftment and CD45RO memory cells that could maintain T cell activity against infections and tumors, but unfortunately, this strategy has not produced encouraging results so far [84]. Recent studies focused on T cell subsets with the antileukemic effect of the graft while decreasing the rate of GVHD. Most peripheral T cells express the α/β T cell receptor and are the main drivers of GVHD. In contrast, γ/δ T cells do not recognize alloantigens and, therefore, do not contribute to GVHD but show strong cytotoxic activity against hematologic malignancies. Studies have found a positive correlation between increased donor-derived γ/δ T cells and survival in patients undergoing haploidentical allo-HSCT or partially mismatched allo-HSCT. Indeed, selective depletion of α/β T cells preserves graft-versus-leukemia (GVL) and anti-infectious immunity through the preservation of natural killer and γ/δ T cells; this approach has largely been tested in children, showing an outstanding graft-relapse-free survival [85]. Preliminary data in adults seem to confirm the efficacy of this approach in achieving a significantly lower GVHD rate, preserving the GVL effect and without a relevant increase in infections [86].

Recent evolution of the T-depleted haploidentical HSCT [85] requires the megadose of CD34+, added by 1 million conventional CD3+ cells/kg (Tcons) under the protection of 2 million/kg Tregs, infused 4 days before, based on the experimental model showing that the prevention of GVHD correlates with the early administration of Tregs [87]. These strategies do not generally employ post-transplant pharmacologic GVHD prophylaxis.

Blockade of CD28 co-stimulatory domain with specific CTLA-4Ig inhibitor may also be effective in preventing GVHD; T cell activation requires at least two signals: the first is triggered by antigen-specific TCR engagement of antigen-MHC and the second is provided by CD28 binding to B7 ligands (CD80 or CD86) so engaging APC; T cells receiving the first but not second co-stimulatory signal, acquire antigen-specific anergy. In experimental models, anti-CD28 antibodies prevented aGVHD, allowing CTLA-4/B7 ligand co-inhibitory function to be unopposed by co-stimulatory CD28/B7 ligand engagement [88]. Abatacept (ABA), inducing a CD28:CD80/86 co-stimulation blockade, has shown to be effective in controlling early T cell allo-proliferative escape; the ABA2 Study compared Standard Prophylaxis (SP) including CNI plus MTX, versus Abatacept + SP, in two different settings: 8/8-HLA-matched unrelated donor in an RCT, while in the 7/8-HLA-mismatched setting, Abatacept was compared with a historical CIBMTR registry series of controls: a significant reduction of aGVHD incidence was observed in both the two setting, and this translated in a significantly better NRM, particularly in the 7/8 setting [89].

CCR5 is a chemokine receptor involved in GVHD pathogenesis as clearly demonstrated in murine models [90,91]; Maraviroc, a CCR5 antagonist, appeared promising in the reduction of severe aGVHD in the liver and gut in a phase 2 clinical study [92]. However, a recent randomized clinical trial showed no advantage with Maraviroc’s addition to the conventional GVHD prophylaxis [93].

Vedolizumab is a humanized monoclonal antibody that targets α4β7 integrin expressed on T lymphocytes and inhibits their gastrointestinal trafficking and adhesion on gut endothelial cells. Moreover, α4β7 integrin was upregulated on the surface of naive and memory T cells in patients who subsequently developed intestinal aGVHD after allo-HSCT [94]. In a large RCT conducted in 333 patients receiving HSCT, the addition of Vedolizumab to a CNI with MTX or MMF resulted in a higher lower intestinal aGVHD-free survival by day +180 (85.5% versus 70.9%). Vedolizumab was also associated with a lower incidence of lower GI aGVHD (7.1% versus 18.8% in the control arm) with a favorable safety profile [35,95].

Recent preliminary findings suggest microbiota modulation and probiotics can induce regulatory T cells and reduce the risk of severe aGVHD following allogeneic hematopoietic stem cell transplantation [96].

Other agents utilized for aGVHD prophylaxis include anti-IL2 R antibodies, but with uncertain results [97,98]. Recently, a Chinese group explored the cGVHD prophylactic efficacy of repeated infusions of mesenchymal stem cells (MSCs) starting 100 days after haplo-HSCT, which significantly decreased the incidence and severity of cGVHD in a randomized clinical trial (27.4% vs. 49.0%; *p* = 0.02) [99]. Table 3 shows the main current and developmental approaches for minimizing the risk of acute and cGVHD.

## 4. Therapy of aGVHD

The treatment intensity of aGVHD strictly depends on the severity of the organ involvement. The aGVHD severity has been traditionally graded (grades 0–IV) by the pattern of organ involvement using the classic Glucksberg–Seattle criteria (GSC) [100]. The GSC classification staged skin, lower GIT and liver, and the combination of these organ-specific grading allowed the create an overall grade of I (mild) to IV (life-threatening), where the overall aGvHD grade typically corresponds to the highest grade conferred by the individual staging of each organ. The Keystone aGVHD consensus panel confirmed the predictive value of the maximum aGVHD grade for day 100 mortality [101]. Further refinements of these criteria have been proposed: the International Bone Marrow Transplant Registry [102] proposed a Severity Index that seems more predictive of TR. In contrast, the grading proposed by the Minnesota group limits the overall grade IV aGvHD to skin and gut stage IV instead of skin and liver stage IV as proposed in the Keystone criteria [103]. Recently, the Mount Sinai Acute GvHD International Consortium (MAGIC) revised these criteria [2], and today, these criteria are the current standard to diagnose and score the severity of aGvHD, especially as regards the upper GIT symptoms and stage 4 skin and GIT involvement (Table 4).

The standard first-line treatment for grades II–IV aGVHD is still represented by high-dose systemic corticosteroids (typically prednisone at 2 mg/kg/day). However, aGVHD fails to respond to steroids in approximately 30–50% of patients; moreover, fewer than half of patients achieve durable Complete Response (CR) after first-line steroid treatment [103]. The commonly used timelines for evaluating the Overall Response Rate (ORR) are days 28 and 56 after the start of treatment, but an earlier evaluation is strongly encouraged in severe cases. A minimal or absent response to first-line steroids is the commonly accepted definition of steroid refractoriness (SR-aGVHD); moreover, we commonly define as SR-aGVHD also the inability to maintain aGVHD control upon tapering corticosteroid therapy. The outcome of patients with SR-aGVHD is dismal, with up to 60–85% incidence of NRM at 2 years, partly due to aGVHD but also to the increased rate of infections. Relapse of the underlying malignancy, facilitated by the immunosuppressive therapy, also concours to NRM. In summary, only 35% of patients with aGVHD achieve the CR by day 28 with steroids alone, and those who do not respond by day 28 experience around 50% of NRM at 6 months, compared to 15% NRM in those achieving CR [104,105].

Several mechanisms of steroid resistance have been identified, including the refractoriness of non-immune effectors (macrophages, polymorphonuclear cells, natural killer cells) involved in the final organ damage, but also Th17 cells are often intrinsically steroid resistant [106]. Clinical manifestations of transplant-associated microangiopathy (TAM) are also associated with SR-aGVHD; whether this is an epiphenomenon or the consequence of direct damage is a matter to debate. However, endothelial damage markers such as Angiopoietin2 and Suppressor Tumor 2 (ST2) are frequently found in patients with TAM and concomitant SR-aGVHD [107].

Many centers treat mild aGVHD of the skin (grade I) with topical steroids alone, but if the rash worsens or new organs are involved, systemic steroids are recommended; patients developing isolated upper GIT aGVHD are characterized by high response rate with Prednisone (PDN) at 1mg/kg with/without non-absorbable steroid [108].

Mielcarek conducted a phase 3 study [109] to test the hypothesis that initial therapy with a lower dose of PDN could be effective for patients with newly diagnosed aGVHD. Patients with grade IIa manifestations (upper gastrointestinal symptoms, stool volumes <1.0 L/day, rash involving <50% of the body surface, without hepatic dysfunction) were randomized to receive PDN at 1 mg/kg/day or 0.5 mg/kg/day. Those with grade IIb or higher manifestations (rash involving ≥50% of the body surface, stool volumes ≥1.0 L/day or hepatic involvement) were randomized to receive PDN at 2 mg/kg/day or 1 mg/kg/day. With a median follow-up of 36 months, the lower dose PDN was effective for patients with grade IIa manifestations since it did not increase the likelihood of secondary immunosuppressive therapy, but in patients with skin-predominant grade IIb or higher manifestations, the lower PDN at 1mg/kg/day was associated with an increased risk of requiring secondary immunosuppressive therapy (41% versus 7%).

aGVHD of GIT presents both diagnostic and therapeutic-specific challenges (Figure 3). These patients should be carefully monitored for stool daily volume and to rule out infective agents such as Clostridium Difficile, Campylobacter, Cytomegalovirus (CMV), Adenovirus or Enterovirus. Upper endoscopy and flex sigmoidoscopy are strongly recommended [110]. However, the diagnosis of GIT aGVHD is based on clinical criteria and in those patients with suspected grade II–IV GIT aGVHD, it is recommended not to wait for histological confirmation and to check the eligibility of the patient for a clinical trial; if not, PDN at 2 mg/kg/day still remains the standard of care. Response to first-line treatment should be monitored daily: in case of worsening after 72 h or in the absence of improvement after 5–7 days, we suggest boosting the treatment (and checking again for clinical trial availability). In the case of CR, a quick steroid tapering (or a slow tapering, in case of partial response, PR) is encouraged; in our personal view, we consider the early association of Extracorporeal Photopheresis (ECP) in order to allow an easier steroid tapering, preventing possible flares and reducing the risk of infections [111].

The MAGIC consortium validated an algorithm based on serum concentrations of two biomarkers, ST2 and regenerating islet-derived protein 3-a (REG3a), to generate a probability score for predicting NRM. The score generated by this algorithm after 1 week of treatment separated steroid-resistant patients into two groups with dramatically different non-relapse mortality (NRM) and survival (OS); the initial response to systemic steroids correlated with day + 28 response, with 1-year NRM and with OS [105].

## 5. Treatment of SR-aGVHD

ECP is widely used in patients with SR-aGVHD being characterized by pleiotropic activity, in particular, the ability to generate tolerogenic dendritic cells [112]; the recognition of photosensitized apoptotic cells, induces an anti-inflammatory response with downregulation of pro-inflammatory cytokines and higher production of anti-inflammatory cytokines; moreover, the induction of tolerogenic APC promotes generation and expansion of regulatory T cells [111]. A meta-analysis of six prospective clinical trials evaluating ECP in SR-aGVHD reported pooled response rates for skin, liver and gut SR-aGVHD of 86%, 60% and 68%, respectively [113]. In a small series of patients with GIT and liver SR-aGVHD, Alemtuzumab showed a promising response rate, although in many cases, this treatment has been complicated by severe infections [114].

Janus kinases are intracellular tyrosine kinases involved in cytokine and growth factor-mediated signal transduction through the activator of transcription (JAK-STAT); JAKs signaling is involved in all three phases of the pathogenesis of aGVHD. In a murine model, JAK1/JAK2 inhibition impaired the proliferation of effector T cells, suppressed pro-inflammatory cytokines, decreased histopathological GVHD grade, and improved OS of mice with aGVHD [115]. Ruxolitinib is a selective inhibitor of the JAK pathway, which is crucial for the release of inflammatory cytokines and subsequent activation of APC, affecting the priming and activation of alloreactive T cells [116]. A phase 3 clinical trial led to the approval of Ruxolitinib for SR-aGvHD in patients ≥12 years by the Food and Drugs Administration (FDA) in 2019 and by the European Medicines Agency (EMA) in 2022; Ruxolitinib was compared with the best available treatment (BAT) in 309 patients with SR-aGVHD; the primary endpoint was the overall response rate at day 28, achieving a significantly better ORR response in the Ruxolitinib arm (62% versus 39%); the durable ORR persisted significantly better in the Ruxolitinib arm at day 56:40% versus 22%; the median Failure Free Survival (FFS) was significantly longer with Ruxolitinib, but still disappointing (5 months versus 1 month) and the median OS was 11.1 months in the Ruxolitinib arm, compared to 6.5 months in the control group [117].

Current investigational approaches in Ruxolitinib-resistant aGVHD (defined as progression of GVHD after at least 5 to 10 days of treatment with Ruxolitinib or lack of improvement in GVHD after ≥14 days of treatment with Ruxolitinib or loss of response) [118] include several molecules such as Itacitinib, another JAK-inhibitor which unfortunately failed to be superior to Ruxolitinib in the Gravitas study, at least in the setting of newly diagnosed aGVHD [119,120] 28 August 2024 8:49:00 A.M. Tocilizumab, an IL-6 receptor antagonist that increases regulatory T cells and decreases pro-inflammatory T cells, showed promising but very preliminary results in a few cases [121,122]. A1antitrypsin prevents organ damage by inhibiting neutrophil elastase and shows immunomodulatory functions, suppressing pro-inflammatory cytokines and inducing regulatory T cells. In a phase 2 trial including 40 adults with SR-aGVHD, an ORR of 65% has been reported with this drug [123].

Three small series of patients with SRaGVHD have been treated with Begelomab, a monoclonal antibody targeting CD26 on CD4+ T lymphocytes, for a total of 69 patients: day 28 response was 75% in the prospective studies and 61% in the compassionate use. Overall, there were 64%, 56%, and 68% responses for skin, liver, and gut stage III–IV GvHD. The OS at 1 year was 50% for the prospective studies and 33% for the compassionate use patients [124].

Besides immunosuppressive drugs, the regeneration of host tissues may be crucial in patients with profound tissue damage, particularly of the GIT; lithium promoted intestinal repair in patients with denuded mucosa [125], and IL-22 restored regenerating islet-derived protein 3, after Paneth cell destruction and facilitated regeneration of GIT epithelium [126,127].

Reducing dysbiosis of the gut microbiome with fecal microbiota transplant (FMT) showed promising results in SR-aGVHD. In the prospective Heracles trial, 24 patients with severe SR-aGVHD received FMT with the MAAT013, a highly diverse multi-donor FMT product manufactured by Maat Pharma, achieving an overall GIT response rate of 38%; in a further 52 patients treated in a French early access program, the ORR was 58% with a 12 months OS of about one-third of FMT treated patients [128].

Mesenchymal Stem Cells (MSC) are characterized by pleiotropic immunomodulatory and anti-inflammatory effects, contributing to tissue repair and immunomodulation [129,130]. Third-party donors MSCs induced immune tolerance [131] and production of anti-inflammatory cytokines, such as TGF-β and IL-10, when exposed to a pro-inflammatory milieu [132,133]. MSCs can be obtained from peripheral blood or bone marrow and the adipose tissue, umbilical cord, amniotic membrane and placenta, potentially interesting MSC sources [134].

Preliminary experiences in adults confirmed the safety of MSC use, and after the early studies showed an encouraging response to MSCs in SR-aGVHD [135], several clinical trials tested MSCs in aGVHD. Up to today, a relevant number of trials with MSC have been undertaken or are ongoing both in acute and in cGVHD settings, but despite several reports of positive outcomes from the use of MSCs for treating aGvHD, the evidence to date from RCT is still lacking [136]. A recent systematic review found that the best candidates for achieving a benefit from MSC therapy are the patients younger than 10 years old, with preeminent skin GVHD but not liver involvement, with a higher naïve T, B cell count and better thymic function. Furthermore, MSCs from younger donors, multiple infusions of MSCs, and prompt treatment before SR develops were correlated with maximum clinical benefits [136]. However, there was extensive heterogeneity and no standardized approach in these clinical trials for several MSCs-donor-related factors, MSCs-related characteristics and clinical-trial design. All these factors may have an impact on the potential efficacy of the MSC product.

Recently, a commercial (Remestemcel) human BM-derived MSC product showed a favorable safety profile in clinical studies and, based on some promising data from a single-arm study in a treatment-naive aGVHD pediatric population [137], it was approved for use in Canada and New Zealand for SR-aGVHD in pediatric patients [138]. Remestemcel was not approved for use by the FDA as it failed to show significant superiority over the placebo in a randomized, phase 3 study in patients with SR-aGVHD [139]. Notwithstanding, a post-hoc analysis of this phase 3 trial in patients with liver GVHD involvement showed a significantly higher durable response rate (29% versus 5%; *p* = 0.047).

Several reasons could explain the disappointing results of MSC in the randomized trials conducted in aGVHD: 1—the significant variability of biological activity of these MSC reflecting the product such as donor age, tissue source and manufacturing process or the medium composition; 2—patient-related factors, particularly the age and involved organs; 3—as for the clinical trial design: the timing, dosage and frequency of MSCs administered. Finally, in some experiences, MSCs were administered with other GVHD therapies, making it challenging to determine MSCs’ benefits. In order to achieve more reproducible results, a phase 3 study (MC0518; Medac GmBH, Theaterstrasse 6, 22880, Wedel Germany) has been planned and is currently ongoing in SR-aGVHD; the MSC product used in this study has been obtained not from a single donor but by pooling the BM cells of unrelated, healthy donors in order to standardize its activity better. Preliminary data showed the safety and the promising activity of MC0518 administered in 69 patients with refractory grade II–IV aGVHD, with 83% ORR at day 28 and 71% OS at 6 months [140,141].

This kind of standardized approach in regards to donor-related factors, also including the evaluation of MSCs-related characteristics, as well as clinical trial design, to limit heterogeneity in trials for aGVHD and to fulfill the requirements of regulatory agencies, has been recently proposed by the International Society for Cell and Gene Therapy Clinical Translation Committee [142].

Finally, the possible use of placenta-derived decidual cells as an alternative source of immunomodulating cells instead of the MSC (although clinical data about their efficacy are still lacking) seems attractive because this source is affected by minor ethical issues [143].

## 6. Treatment of cGVHD

Like aGVHD, the cGVHD treatment is also modulated according to the severity score of the disease.

The 2005 National Institute of Health (NIH) consensus proposed a scoring algorithm for evaluating the overall severity (NIH global score) of cGVHD, which in turn was based on the combination of the severity of manifestations in eight individual organs (NIH organ severity score, including skin, lung, liver, joints upper and lower GIT, mouth, eye) [3]. The broad scoring categories help to classify patients into three categories (mild, moderate and severe score) and provide clinically meaningful information about the disease extent and severity; cGvHD global severity scoring is a powerful prognostic tool, and the current use of this NIH severity score strongly predicts the survival and the association between organ severity and the risk of mortality, seems to differ according to individual organ sites [4]. The strongest associations were observed for the lung, followed by the liver, skin, gastrointestinal tract and mouth [144].

Table 5 shows the current cGVHD severity score compared to the previous classifications by Shulman [145].

There is general agreement that systemic treatment should be reserved only for moderate to severe cGVHD forms, while local treatment is generally reserved for mild GVHD forms.

The main goals of treatment for cGVHD are: i—to reduce symptom burden; ii—control objective manifestations; iii—prevent progression of disease activity; iiii—preserve irreversible organ damage; iiii—improve quality of life and the OS, achieving eventually the withdrawal of systemic immunosuppressive treatment.

The current first-line standard for cGVHD is still PDN, generally at 0.5–1 mg/kg per day, often in combination with a CNI. However, long-term treatment with corticosteroids is associated with several serious side effects. Furthermore, high failure rates have been observed in patients with cGVHD receiving first-line treatment with corticosteroids, alone or combined with other treatments. Consequently, there is an unmet medical need for new first-line steroid-free treatments for cGVHD. Up to today, unfortunately, all RCTs were unable to demonstrate the superiority for any investigational treatment, compared to steroids alone [127,146,147,148,149]; indeed, the FDA has not approved any drugs or devices for first-line therapy of moderate to severe cGVHD; however, steroids cannot be accepted as a golden standard for initial treatment of cGVHD and prospective clinical trials, including steroid-free approaches are currently encouraged [150].

## 7. SR-cGVHD

ECP is an established, clinically effective second-line therapy for SR cGvHD; ECP has been used in over 70,000 patients, confirming its excellent safety profile, and a meta-analysis showed a pooled ORR of 68% in patients with SR-cGVHD [151]. ECP has a pleiotropic mechanism of action and should be considered as an immunomodulating therapy rather than an immunosuppressive treatment; its immunomodulatory effect is antigen-specific, and the reinfusion of apoptotic cells after UVA irradiation leads to phagocytosis by APCs, increased production of anti-inflammatory cytokines, modulation of T cells toward a Th2 phenotype, maturation of tolerogenic DCs and promotion of Treg cell generation, which play a pivotal role in contrasting alloreactive T-cells and aberrant B-cells [111]. A recent retrospective study showed that combining ECP with Ruxolitinib was safe and effective in a small group of patients with SR-cGVHD [152].

Today, the FDA has approved three drugs for SR-cGVHD: the Jak/Stat inhibitor Ruxolitinib, the BTK inhibitor Ibrutinib and Belumosudil, a Rock2 inhibitor [153,154].

Besides its strong activity in aGVHD, Ruxolitinib showed high activity in cGVHD, promoting increased levels of FoxP3+ regulatory T cells, which play a role in immune tolerance; moreover, preclinical evidence suggests that the development and activation of B cells may be regulated by JAK signaling and that also the macrophages and the fibroblast activation may be inhibited through JAK-2 inhibition [155]. Zeiser compared in an RCT Ruxolitinib with BAT in 329 adults with SR-cGVHD: the ORR at week 24 (the primary endpoint) doubled in the Ruxolitinib arm (49.7% versus 25.6%); moreover, the Ruxolitinib led to a significantly longer median FFS (18.6 months versus 5.7 months) [156].

ROCK2 signal is critical in regulating pro-inflammatory cytokines such as IL-21 and IL-17, which play a central role in the pathogenesis of cGVHD [157]; Belumosudil (a potent ROCK2 inhibitor) reduces Tfh cells via downregulation of STAT3 and enhances T reg via STAT5 [158]. ROCK2 also represents a pivotal profibrotic signal by regulating fibroblast differentiation to myofibroblasts; moreover, Belumosudil showed a strong reduction of lung and skin fibrosis in Bronchiolitis Obliterans Syndrome (BOS) animal models [158,159]. A recent trial confirmed the clinical efficacy of Belumosudil in multirefractory cGVHD. Two different dosages of Belumosudil were compared, achieving a similar response of 74% and 77% of ORR, respectively [160]. A limitation was that all subjects received Belumosudil; indeed, requiring randomization to BAT was not deemed ethically appropriate because all the patients had already attempted a median of three prior lines of BAT with ECP, Ibrutinib and Ruxolitinib (29%). According to the NIH response criteria [161], the best ORR for lung cGVHD was 32% (PR: 17%; CR: 15%) [162].

Increased BAFF levels are associated with circulating, activated B cells in cGVHD patients; these activated B cells have increased survival capacity through both BAFF-associated and BCR signaling pathways [163]. Taking lessons from B cell malignant diseases, some researchers tested the activity of monoclonal antibodies selectively able to deplete B cells; in SR-cGVHD, the anti-CD20 Rituximab conferred some clinical benefit [164], but with modest long-term success, likely due to low or absent CD20 expression in plasma cells.

Ibrutinib blocks BCR, T-cell Bruton tyrosine kinase (BTK) and IL2 inducible tyrosine kinase (ITK). Preclinical models receiving HSCT from BTK or ITK-deficient donors did not develop cGVHD, suggesting a possible role for Ibrutinib for the treatment of cGVHD; Ibrutinib has been tested in a phase 2 study in a small group of 42 SR-cGVHD patients: the best ORR was 67% and 71% of the responders showed a sustained response for 20 weeks but 33% stopped Ibrutinib, due to adverse events [165].

Other compounds with specific anti-B cell activity have been tested: Syk is involved in BCR signaling, and increased amounts of Syk have been observed in patients with cGVHD. Syk inhibitors such as Entosplenib successfully reversed manifestations of cGVHD in a murine model of bronchiolitis obliterans [40,166].

The immune ablation by T memory cell depletion has been successfully tested in some refractory autoimmune diseases, achieving a “reset” of the aberrant immune system, and this proof of principle has also been reproduced in two pediatric patients with severe SR-cGVHD, achieving the immune ablation” by using the combination of Cyclophosphamide, Fludarabine and ATG, followed by reinfusion of purified CD34+, obtained after mobilization with Cyclophosphamide [167].

A novel approach in SR-cGVHD is represented by some promising compounds targeting non-immune effectors. Blocking the exaggerated collagen production is an attractive tool: TKI is a potent dual inhibitor of PDGF-R and TGF-b pathways. We tested Imatinib in 39 patients with multi-refractory cGVHD, with fibrotic features achieving a 51% ORR and a 46% EFS at 3 years [168]. In experimental models, Nilotinib demonstrated a pleiotropic activity in reducing inflammatory cytokines such as TNF and IL-17 and showed a potent anti-TGF-beta inhibition; Nilotinib exposure switches off the SMAD signal in fibroblasts, translating in a marked improvement of the fibrotic lesions in treated patients [45,46]. Nintedanib and Pirfenidone have been evaluated in fibrotic lung diseases and showed promising activity also in fibrotic lung GVHD [169,170].

Macrophage depletion by using an anti-CSF-1R monoclonal antibody was also effective in cGVHD. Axatilimab is a humanized IgG4 monoclonal antibody recognizing the CSF-1R, inhibiting monocyte activation. In a phase 1–2 study, Axatilimab was tested in 40 patients with SR-cGVHD. The best ORR observed at any point was 69%, with 33% of patients experiencing sustained response lasting 20 weeks [171]. These data indirectly suggest that targeting profibrotic macrophages could be a promising novel strategy with a favorable safety profile for SR-cGVHD.

Finally, strategies augmenting regulatory mechanisms are being applied for cGVHD prevention and cGVHD therapy [172]. CD4+CD25 high Tregs express the master regulator transcription factor FOXP3, which is critical for their survival and suppressor functions [173]. Thymic Tregs have a self-antigen-specific TCR repertoire [173]; in contrast, peripheral Tregs are differentiated from mature CD4+ T cells extrathymically in response to self-antigens and foreign antigens in the context of immunoregulatory molecules such as retinoic acid and TGF-β [173]. Tregs suppress T cell responses and can rebalance the peripheral tolerance, achieving the GVHD control. The adoptive transfer of ex vivo expanded donor or third-party Tregs was demonstrated to suppress aGVHD [174]. Most studies currently isolate human Tregs from peripheral blood, although third-party umbilical cord blood-derived Tregs have also been successfully used in clinical trials [175].

## 8. Management of Patients with Lung Involvement

Bronchiolitis Obliterans Syndrome (BOS) is currently the only entity recognized as a diagnostic manifestation of pulmonary cGvHD [4]. It affects up to 14% of patients with cGvHD; it usually develops between 100 days and 2 years of HSCT, but late onset beyond 5 years post-HSCT has been reported [176]. Recognized risk factors for cGVHD-BOS are impaired lung function before HSCT, a myeloablative/busulfan-containing conditioning regimen, CMV seropositivity, pre-transplant history of pulmonary disease, female donor, unrelated donor and prior aGVHD, while previous ATG prophylaxis reduces the risk of BOS [177]. Similarly to chronic lung allograft dysfunction (CLAD) after lung transplantation [178], other clinical phenotypes of lung cGVHD have been identified, including less defined restrictive or mixed phenotype patterns characterized by interstitial lung disease or atypical radiological pictures [179,180]. The pathology of GVHD-BOS is less well-defined than CLAD-associated BOS due to a lower number of lung biopsies and lower autopsy rates [181]. Two distinct patterns of BOS have been described: constrictive bronchiolitis obliterans (CBO) and lymphocytic bronchiolitis [182]. CBO demonstrates marked bronchiolar narrowing with fibrous lesions and hyperplasia of the epithelium, whereas in lymphocytic bronchiolitis, fibrosis is absent, and there is bronchiolar dilation and epithelial thinning, necrosis or disappearance. Patients with lymphocytic bronchiolitis have better survival and better response rates for steroids than those with CBO [182,183]. The diagnosis of lung cGVHD is often difficult; outside the classic BOS diagnosed according to the NIH criteria, non-BOS lung GVHD (generally characterized by restrictive dysfunction) requires a complex work-up to exclude non-lung complications (Figure 4 Management of Lung GVHD).

Traditionally, treatment options for BOS have consisted of standard cGVHD therapies, including systemic corticosteroids and immunosuppressive agents; today, there is limited evidence to guide a specific treatment of BOS. The European Society for Blood and Marrow Transplantation recommends the combination of Fluticasone, Azithromycin and Montelukast (FAM) with a steroid pulse and rapid taper over 1 month [184]. This recommendation was based on data from a small non-randomized study in 36 patients with BOS after HSCT who received FAM plus a steroid burst. Other immunosuppressive or immunomodulatory therapeutics can halt disease progression but rarely improve pulmonary function or symptoms. Therefore, effective therapies for BOS remain an unmet need. Outside of Ruxolitinib, which has been approved for second-line cGVHD treatment but for which there is modest evidence of specific activity in BOS, there are no standard second-line treatments for lung cGVHD, albeit some retrospective data suggest that ECP could improve survival [185]. One trial in a small population of steroid-refractory lung cGVHD evaluated the use of a TNF-α inhibitor, Etanercept, and showed LFT improvement (i.e., ≥10% increase in FEV1) in one-third of patients with BOS [186].

Preliminary data suggest that Belumosudil could be effective in patients with BOS after the failure of several lines of treatment, including Ibrutinib and Ruxolitinib [160]; a post-hoc analysis in 59 BOS patients documented an improvement in FEV1 of ≥10% in 22% of patients; the best ORR, based on lung function was 24%, with 82% 2-year OS [162].

Up to today, several compounds targeting non-immune effectors have been explored in lung SR-cGVHD; among them, Alvelestat could be effective in blocking the damage induced by PMN in BOS; this neutrophil elastase inhibitor showed anti-inflammatory properties in human alveolar and bronchial epithelial cells by targeting neutrophil extracellular traps and can improve survival rates in mice models [187]. Based on their antifibrotic activity in preclinical models and according to preliminary clinical trial results, TKI and Nintedanib have currently been tested both in fibrotic lung diseases and in fibrotic lung GVHD with promising results [168,169,170]. Today, the treatment of lung GVHD still represents a challenge, with a scenario of options not only including immunomodulatory or anti-inflammatory agents but also supportive care, including infection treatment and prevention with a reasoned use of vaccination and immunoglobulins. Finally, the role of rehabilitation is crucial; also, continuous oxygen therapy in severely compromised patients and lung transplantation in selected cases should be considered [188].

The onset of dyspnea (with or without other symptoms) and/or the deterioration of the Lung Functional Test (LFT) require a careful work-up based on both imaging and bronchoalveolar lavage, to exclude infectious complications is recommended; DLCO testing, although not mandatory, is helpful since its reduction is frequent in restrictive syndromes. An isolated reduction in FEV1 (or FEV1/VC ratio) of at least 10% over 3 months, if associated with distinctive radiological findings (bronchiectasis, air trapping, bronchial wall thickening) points towards typical lung GVHD (BOS). Alteration of the LFTs, but with a preserved FEV1/VC ratio (often with reduced DLCO), instead points towards a restrictive syndrome or a mixed form (which may be referred to as an atypical form of cGVHD). In this case (in the absence of microbiological/virological isolations), radiological imaging can be indicative of either rare forms (PPFE, Pleuroparenchymal fibroelastosis; NSIP, Non-Specific Interstitial Pneumonia) or may show consolidations indicative of COP (Cryptogenic organizing pneumonia). Steroid treatment can be rapidly curative in COP while it rarely is in BOS; in the case of SR-cGVHD or steroid-dependence, the association of ECP for 10–12 months is an option; Ruxolitinib is indicated in any case in SR-cGVHD forms, while in Ruxolitinib-refractory forms, consider the use of Belumosudil (if available); other options include the use of Belumosudil, Imatinib and Ibrutinib and enrollment clinical trials. Ancillary treatment with FAM is generally recommended in BOS forms, while in young patients with severe impairment of lung function, the option of a lung transplant should be considered.

## 9. Management and Topical Treatment of Specific cGVHD Localizations (Skin, Oral, Ocular and Genital cGVHD)

Skin is the most commonly involved organ in cGVHD, occurring in approximately 75% of cGVHD patients [189]. Cutaneous manifestations of cGVHD are associated with pruritus and pain, limited range of motion, and increased risk of wound infection; skin cancer incidence is also increased in patients with cGVHD. Ancillary care of the skin includes the management of symptoms such as pruritus, rash and pain, as well as the topical care of ulcerations and superinfection. Skin-directed therapies may improve skin disease control and quality of life without causing the adverse effects of systemic immunosuppressive treatment.

Among the risk factors, environmental ultraviolet (UVA; UVB) radiation is important, causing exacerbation of cutaneous cGVHD [190]. Therefore, photoprotection is mandatory with clothes and topical agents. Topical steroids and emollients can improve cutaneous cGVHD, particularly the non-sclerotic skin lesions (e.g., lichen-planus-like or papulosquamous plaques), but long-term topical steroids may induce local skin atrophy. Phototherapy is another option for patients with extended skin cGVHD involvement, as well as for those who fail to respond to topical treatment or who have become steroid-dependent [191].

Oral cGVHD involvement, including the salivary glands, is frequent (up to 80%) and represents a major cause of functional impairment. Lichenoid changes are the most common feature, followed by erythema and ulcerations. Oral cGVHD often requires ancillary therapy despite the good responses in other sites of cGVHD, and regular long-term follow-up is recommended due to the risk of oral squamous cell carcinoma [192]. Like skin manifestations, the Consensus Development Project on Criteria for Clinical Trials in Chronic Graft-versus-Host Disease proposed some clinical guidelines for managing oral cGVHD [193]. Topical corticosteroid preparations are the mainstay; topical application of CSA or azathioprine has also been tested in lichenoid manifestations, in association with systemic immunosuppressive therapy, in patients with ulcerative lesions resistant to topical steroids [194]. In a randomized clinical trial, intensive application of topical Dexamethasone at 1 mg/mL solution, rinsed for 1 month intensively 3 to 4 times a day for 5 min, was safe and effective, while Tacrolimus, 1 mg/mL solution, albeit well tolerated, appeared less effective [192]. Ancillary care for dry mouth may include frequent water sipping, saliva stimulants (e.g., sugar-free gum), oral moisturizing agents, and saliva substitutes.

The clinical spectrum of ocular cGVHD (oGVHD) includes acute conjunctival inflammation, cicatricial conjunctivitis, and, most frequently, keratoconjunctivitis; corneal involvement is quite common with up to 66% of cases. The primary aim in oGVHD management is to preserve vision and quality of life by improving lubrification of the ocular surface, reducing ocular surface inflammation and preserving corneal epithelium integrity; topical steroids have been used for cicatricial conjunctivitis in limited case series. For lubrication, the range of adjunctive measures includes the use of preservative-free artificial tears to coat the ocular surface; oral muscarinic agonists (e.g., pilocarpine) can improve tear production, particularly in those patients with scleroderma-like manifestations and lacrimal (or ductal) gland damage. Autologous serum eye drops, characterized by anti-inflammatory and regenerative properties, could also be effective. Amniotic membrane transplantation has been demonstrated to be effective for those patients with severe disease showing corneal perforation and symblepharon [195]. Finally, the use of scleral lenses in patients with severe ocular GVHD has been shown to reduce ocular pain and improve visual acuity [196].

Genital cGVHD affects about 25% of women undergoing HSCT, presenting with abnormalities of the mucosa or sclerotic changes. Common symptoms include dysuria, dryness, and dyspareunia [197]. Typically, the vulva shows erythematous patches, lichen-planus-like lesions and agglutination of vulvar structures with adhesions or fibrosis between the clitoris and prepuce, and less often, ulcers; vaginal findings include mucositis and ulcers [198]. Topical ultra-high potency corticosteroid is the mainstay of therapy, but topical calcineurin ointments also have efficacy. The anti-inflammatory antibiotic clindamycin has been used successfully as an intravaginal therapy [199]. Patients may develop candidiasis or experience recurrence of Herpes Simplex Virus (HSV) or Human Papilloma Virus (HPV) during immunosuppressive therapy and must be monitored. The sclerotic features may also require dilators, while surgical lysis with or without vaginal reconstruction is indicated for patients with extensive synechiae and complete obliteration of the vaginal canal [193,200].

## 10. Conclusions

Both acute and cGVHD still represent a significant cause of NRM and deteriorated Quality of Life (QoL). The prevention and treatment of GVHD still remain a challenge; today, despite the substantial improvement in prevention strategies, the incidence of severe forms of aGVHD (grade III–IV) is still relevant, as well as that of severe forms of cGVHD. The extension of indications for HSCT to include elderly subjects has increased the number of subjects at risk of developing GVHD; moreover, the persistently high NRM observed in patients developing severe forms of lower GIT aGVHD and the lung involvement in cGVHD is an unmet need.

The first-line therapy for both aGVHD and cGVHD is still anchored to steroids, as no study has shown significant superiority over this standard. Many patients still develop steroid dependence, with severe adverse effects and potentially life-threatening infectious complications; the persistence of active GVHD, requiring systemic immunosuppressive treatment, causes a state of prolonged immunosuppression while the imbalance of immune subpopulations, with a persistent Treg deficiency, remains a risk factor for self-maintenance of GVHD and the development of flares.

In the setting of acute intestinal GVHD, the main lines of development are based on new knowledge in the field of microbiota and the possibility of stimulating enterocyte regeneration [126], while the possibility of associating Ruxolitinib with a steroid-sparing therapy such as ECP early on could improve the outcome of some high-risk patients [111].

There is still room for improvement in both prevention and treatment strategies. For example, the introduction of Abatacept seems to provide some advantages, mainly in the mismatched transplant setting [201,202]. Also, manipulation strategies, such as the selective depletion of alpha/beta T lymphocytes, seem safe and effective, although expensive.

As for treatment, the possibility of testing the use of standardized MSC cell therapies with the potential of immunomodulating and stimulating the regeneration of damaged intestinal mucosa (in the aGVHD setting) and of counteracting fibrotic remodeling in the context of cGVHD could become a concrete opportunity; the use of expanded Tregs could represent a useful tool both in the prevention and treatment fields although there are still difficulties regarding the optimal sources of these effectors and the modality of their use, which should prevent the phenomenon of their exhaustion in an inflammatory milieu [174].

Using reliable biomarkers in aGVHD could allow for earlier and more aggressive treatment, particularly for those patients with unfavorable risk scores [203].

In the context of cGVHD, an additional challenge is represented by the not-rare clinical scenario of observing, alongside favorable responses in some districts, progression or non-response in others, which suggests the possibility of individualizing and differentiating therapy based on different anatomical targets. Dissecting the acute and cGVHD pathogenesis in the context of specific damage for each organ may be a challenging goal, and preliminary data at the intestinal level [30] suggest some specific checkpoints (e.g., adhesion molecules or microbiota imbalance) that are inevitably different from those of other target organs. The striking similarities between lung cGVHD and CLAD in lung transplant recipients have allowed us to identify non-BOS forms of lung involvement by cGVHD that require further refinement [180,204].

In this context, a new scenario is represented by the availability of new compounds with a mechanism of action that does not directly involve immunological effectors, contrasting the organ damage mediated by monocytes and/or neutrophils [187].

Some new agents can reduce the aberrant activation of fibroblasts and collagen production, opening the possibility to contrast or reverse the fibrotic tissue remodeling, previously judged irreversible [18,32].

It is possible to hypothesize that in the future, the association of biological markers in GVHD patients [203,205] with specific disease manifestations could help us to build new algorithms for specific treatment-related outcomes, refine risk stratification and for the clinical trial design. Moreover, identifying suitable biomarkers could help with the choice of specific therapeutic agents and could be used to monitor responses to treatment.

Today, waiting for the validation of reliable and clinically applicable markers, two possible scenarios could be hypothesized: 1—an organ/clinically driven therapeutic approach (a personalized treatment, i.e., for skin GVHD or gut GVHD or lung GVHD); 2—to explore the association of different compound targeting different pathways, i.e., B cell inhibitors with a non-immune effector inhibitor or ECP combined with another compound (i.e., Ruxolitinib or antifibrotic agents) without overlapping toxicity in order to evaluate a possible synergy.

The need to test new steroid-free or steroid-limited approaches in first-line therapy remains a medical need that should encourage transplant specialists to enroll patients with GVHD in prospective studies.

## Figures and Tables

**Figure 1 cells-13-01524-f001:**
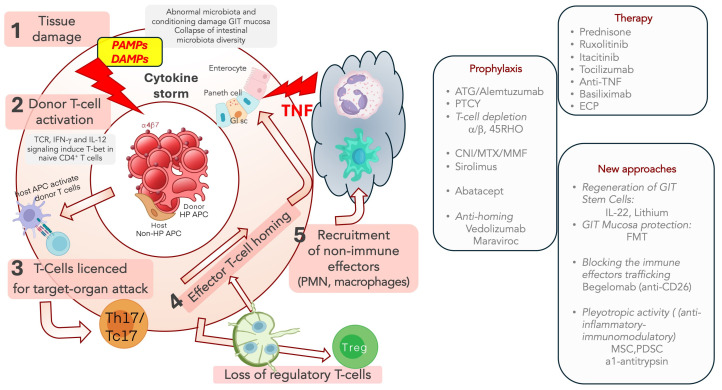
Pathogenesis of aGVHD and main targetable pathways for prophylaxis and treatment. Simplified outline of the aGVHD pathogenesis and the possible specific interventions according to some targetable steps. 1—The conditioning damages host tissues and causes the release of inflammatory mediators with the activation of host APCs; this process is amplified in the GIT lumen, where the altered microbiota and the destruction of the intestinal barrier facilitate the stimulation of innate immunity. 2—Host APCs activate donor T cells in the secondary satellite lymphoid organs 3—After expansion in the lymph nodes, activated T cells are primed, differentiating to type 1 T helper (Th1)/type 1 CD8+ T (Tc1) or Th17/Tc17 cells, and become able to target specific organs; 4—The lymphocyte traffic to the target organs is mediated by adhesion molecules such as L-selectin, CCR7, integrin a4b7; 5—activated T-cells can induce the tissue damage, both directly and by recruiting non-immune effector cells, (such as monocytes, PMN and NK) cells, and cytokines, such as TNF. The progressive loss of Treg contributes to the uncontrolled expansion of alloreactive T cells. Targetable pathways for Prophylaxis: 1—alloreactive T-cell depletion (in vivo/ex vivo): ATG; Campath; 2—early PTCy blocks alloreactive donor T cell expansion; 3—CD34 + selection/a-b depletion; 4—inactivating TCR (CNI: CSA, Tac); Sirolimus is a mTOR inhibitor which inhibits effector T-lymphocytes; 5—Abatacept (CTLA4-Ig) blocks T cell-APC co-stimulation; 6—Anti-homing compounds interfere with the alloreactive T-cell migration in the target organs (Vedolizumab; Maraviroc). Targetable pathways for Therapy: 1—anti-inflammasome treatment (Prednisone; Ruxolitinib/Itacitinib; Tocilizumab; Etanercept/Infliximab; 2—blocking T-cell priming: anti-IL2 (Basiliximab); 3—Primed T cells are susceptible to the Jak1/2 inhibitor Ruxolitinib or anti-IL-6R-antibodies (Tocilizumab) anti-TNF-antibodies; 4—Begelomab (anti-CD26) blocks alloreactive T cell migration to target organs; 5—treatments aimed to protect or regenerate target organs: IL22, FMT, anti-1 anti-trypsin; 6—Agents with pleiotropic activity: MSC; PDSC; multitarget treatments (ECP induces tolerogenic dendritic cells; reduces inflammasome; augments Treg). Abbreviations: aGVHD, acute GVHD; GIT, gastrointestinal tract; GIsc, gastrointestinal stem cells; PAMPs, pathogen-associated molecular patterns; DAMPs, damage-associated molecular patterns; HP APC, hematopoietic antigen-presenting cell; non-HP APC, non-hematopoietic antigen-presenting cell; TCR, T cell receptor; IFN, interferon; IL, interleukin; Th, helper T lymphocyte; Tc, cytotoxic T lymphocyte; T reg; regulatory T cells; PMN, polymorphonucleated cells; NK, natural killer cells; PTCY, post-transplant cyclophosphamide; CNI, Calcineurin Inhibitor; CSA, cyclosporine A; Tac, Tacrolimus; MTX, Methotrexate; MMF, Mycophenolate; ECP, Extracorporeal Photopheresis; FMT, fecal microbial transplant; MSC, mensenchimal stem cell; PDSC, placenta-derived stem cells.

**Figure 2 cells-13-01524-f002:**
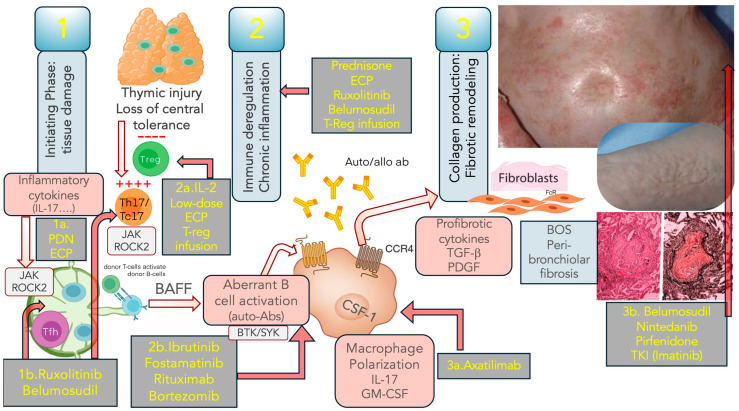
Pathogenesis of cGVHD and main targetable pathways for prophylaxis and treatment. The three-step development of c-GVHD with the treatment targeting the specific pathways. Phase I: Inflammasome is the starting point of cGVHD and could be clinically undetectable; conditioning and aGVHD cause thymus injury with loss of central tolerance. Phase II: T and B cell dysregulation; thymic dysfunction impairs donor Treg generation and the negative selection of autoreactive donor T cell clones that escape into the periphery. Tfh cells expand in lymphoid organs and promote the development of allo/autoreactive B cells in germinal centers; BAFF facilitates survival and expansion of the aberrant B cell clones; the aberrant B cells produce auto or allo-antibodies which induce macrophage polarization in presence of IL17/GM-CSF Phase III: Fibrosis and tissue remodeling; activated macrophages produce high levels of profibrotic cytokines (TGF-b, PDGF) with continuous stimulation of fibroblast which in turn produce exaggerated collagen matrix with tissue fibrotic remodeling. Targetable pathways: 1—The inflammasome and the subsequent phase of deregulation can be targeted by drugs/treatments with anti-inflammatory properties (PDN/ECP/Ruxolitinib); 2—Immune dysregulation and aberrant B-cell activity. The imbalanced Treg/T effector cell reconstitution (and the altered ROCK-2 signal) can be restored by Belumosudil, low-dose IL-2 treatment or by Treg infusions; ECP preserves Treg function. Belumosudil blocks Th17 differentiation and GC reactions by inhibiting Tfh cell generation (like Ruxolitinib); constitutive B cell receptor signaling (BTK) and intracellular downstream Syk can be inhibited by BTK Ibrutinib or Fostamatinib, while Rituximab blocks auto-antibodies production and Bortezomib inhibits long-lived autoreactive plasma cells. 3—Macrophage polarization and fibroblast activation with exaggerated collagen production: fibrosis and tissue remodeling. Axatilimab inhibits CSF1-R+ macrophages secreting profibrotic cytokines; -TGF-β production is inhibited by Belumosudil, Nintedanib, Pirfenidone or by Imatinib; these compounds also inhibit the fibrotic process via PDGF-R. Abbreviations: cGVHD, chronic GVHD; aGVHD, acute GVHD; T reg, regulatory T cell; Tfh, helper T follicular lymphocyte; GC, germinal center; BAFF, B cell activating factor; PDN, prednisone; ECP, Extracorporeal Photopheresis; BTK, Bruton tyrosine kinase; TKI, tyrosine kinase inhibitor.

**Figure 3 cells-13-01524-f003:**
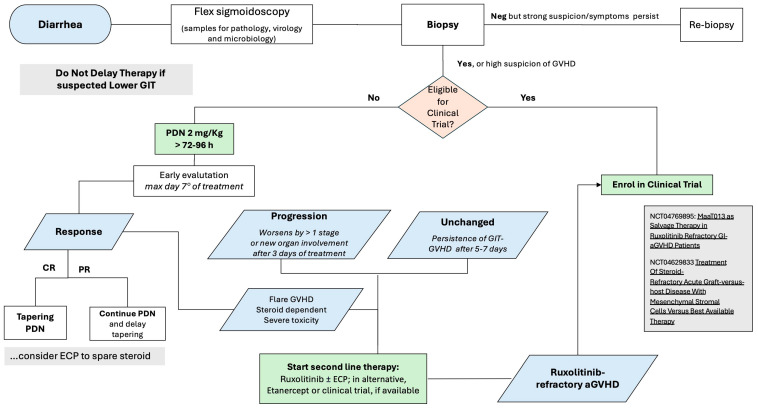
How we manage GIT-GVHD. Proposal of a clinical algorithm for managing patients with grade II–IVlower GIT aGVHD: first of all, do not wait for histologic confirmation in case of suspected GIT GVHD; check the chance to enroll the patient in a clinical trial: if not, start immediately standard first-line therapy with PDN 2 mg/kg and monitor response every day. If worsening after 72 h or without improvement after 5–7 days, a treatment change is strongly suggested (check again for a clinical trial availability). In the case of CR, a quick steroid tapering (or a slow tapering in the case of PR) should be considered; in our personal view, we consider the early ECP association in order to allow an easier steroid tapering, preventing possible flares and reducing the risk of infections. The standard second-line treatment is represented by Ruxolitinib (Etanercept or ECP represents possible alternative treatments), but enrollment in clinical trials (if available) is a valuable option both in steroid-refractory and in Ruxolitinib-refractory aGVHD. Fecal Microbiota Transplant or Mesenchymal Stem cell infusions are currently under evaluation: 1—NCT04769895: MaaT013 as Salvage Therapy in Ruxolitinib Refractory GI-aGVHD Patients; 2—NCT06075706 Treatment Of Steroid-Refractory Acute Graft-versus-host-Disease With Mesenchymal Stromal Cells Versus Best Available. GIT, gastro-intestinal tract; Neg, negative; PDN, Prednisone; CR, complete response; PR, partial response; ECP, Extracorporeal Photopheresis.

**Figure 4 cells-13-01524-f004:**
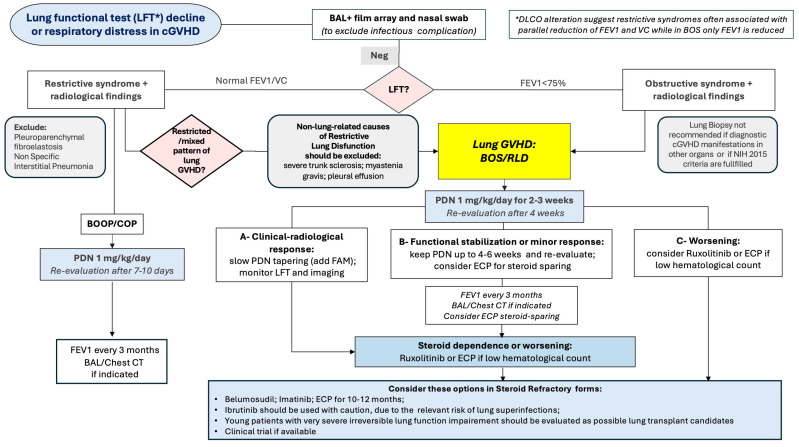
Approach to the patient with suspected pulmonary GVHD involvement: the onset of dyspnea on (with or without other symptoms) and/or the deterioration of the LFTs require a careful work-up based on both imaging and Bronchoalveolar lavage, in order to exclude infectious complications; DLCO testing, although not mandatory, is useful since its reduction is frequent in restrictive syndromes. An isolated reduction in FEV1 (or FEV1/VC ratio) of at least 10% over 3 months, if associated with distinctive imaging findings (bronchiectasis, air trapping; bronchial wall thickening) points towards typical lung GVHD (BOS). Alteration of the LFTs, but with a preserved FEV1/VC ratio (often with reduced DLCO) instead points towards a restrictive syndrome or a mixed form (which may be referred to an atypical form of cGVHD). In this case (in the absence of microbiological/virological isolations) radiological imaging can be indicative of either rare forms (PPFE; NSIP) or may show consolidations indicative of COP. Steroid treatment can be rapidly curative in COP while it rarely is in BOS; in the case of SR-cGVHD or steroid-dependence, the association of ECP for 10-12 months is an option; Ruxolitinib is indicated in any case in SR-cGVHD forms, while in Ruxolitinib-refractory forms, consider the use of Belumosudil (if available); other options include the use of Imatinib and Ibrutinib. Ancillary treatment with FAM is generally recommended in BOS forms, while in young patients with severe impairment of lung function the option of a lung transplant should be considered. Abbreviations: LFT: lung functional test; BAL: Broncho-Alveolar Lavage; DLCO: Diffusion Lung Carbon Monoxide; FEV1: Forced Expiratory Volume in the first second; VC: vital capacity; BOS: Bronchiolitis obliterans syndrome; RLD: Restrictive Lung Disease; PDN: Prednisone; FAM: Fluticasone-Azithromycin-Montelukast; ECP: Extracorporeal Photopheresis; CT: Computed Thomography; PPFE: Pleuropaenchimal fibroelastosis; NSIP: Non-specific interstitial Pneumonia; BOOP: Bronchiolitis obliterans organizing pneumonia; COP: Cryptogenic organizing pneumonia; SR-cGVHD: Steroid-refractory chronic GVHD.

**Table 1 cells-13-01524-t001:** Types of GVHD, based on NIH 2015 definitions and the joint proposal EBMT−NIH−CIBMTR Task Force [7,8].

Type of GVHD	Classical or Late aGVHD	Classic cGVHD	Overlap cGVHD	Indefinite Other cGVHD
Characteristics and Organs involved	Skin: maculopapular rash Liver: elevated bilirubin GIT: anorexia with weight loss, nausea, vomiting, diarrhea, severe pain, GI bleeding and/or ileus	Skin, nails, scalp and body hair Mouth Eyes Esophagus Lungs Muscles, joints and fascia Genitalia	Simultaneous presence of acute and chronic GVHD manifestations	Atypical signs and symptoms of alloreactivity outside NIH diagnostic criteria

**Table 2 cells-13-01524-t002:** Main risk factors for acute and chronic GVHD development.

Risk Factor	aGVHD	cGVHD	References
Stem cell source (PBSC)	Debated Yes	Yes	[10,11,12] [13]
Donor characteristics: Female-to-male graft Older donor Mismatched and unrelated donors	Yes Yes Yes	Yes Yes Not significant	[10,11,14,15,16]
Patient age	Not significant	>12 years	[17]
DLI	Yes	//	[11,15]
Severe infections during the peri-transplant period	Yes	Yes	[10,11,18]
Conditioning (Myeloablative regimen TBI/irradiation-based regimen)	Yes	Yes	[11,15,16,17,18,19]

PBSC, peripheral blood stem cell; DLI, donor lymphocyte infusion; TBI, total body irradiation.

**Table 3 cells-13-01524-t003:** Current prophylaxis strategies.

Strategy/Transplant Setting	Agent/Mechanism	Results	References
Donor selection/MUD	Matching based on high-definition HLA-typing	Mismatches at HLA-DQB1, HDRB3/4/5 increases the risk of aGVHD HLA-B dimorphisms associated with higher GVHD Non-permissive HLA-DPB1 mismatch increases NRM	[48,49,50]
Inactivate alloreactive T-cells MRD/MUD	CNI; MMF Inhibition of TCR Inhibition of alloreactive T-cell proliferation	CyA/MTX: aGVHD II–IV: 44–76%; cGVHD 56% Tac/MTX: aGVHD II–IV: 56% (III–IV 18%); cGVHD 2y 76% Tac/MTX: aGVHD II–IV 31.9% (III–IV 13%); cGVHD 56% Tac/MTX: aGVHD II–IV 74% (III–IV 4%); cGVHD 1y 45% Tac/MMF: aGVHD II–IV 79% (III–IV 19%); cGVHD 1y 38%	[32,53,54] [53] [54] [55]
In vivo T-cell depletion and modulation after HSCT Haploidentical_NMA/MAC Haploidentical_PTCy/Tac/MMF vs. Tac/MTX (RIC/NMA) Haploidentical vs. MUD vs. MRD Haploidentical vs. MUD (RIC/MAC) Haploidentical_BM vs. PBSC (RIC/MAC) Haploidentical_BM vs. PBSC (NMA) Haploidentical MAC vs. RIC_BM vs. PB vs. BM	PTCy Elimination/inhibition of alloreactive T cells; peripheral tolerance; central deletion of alloreactive T cells in the thymus	PTCy with bone marrow source aGVHD II–IV: 34% (III–IV: 6%); cGVHD extensive: 5 *–25% ** (NMA) aGVHD II–IV 17% (III–IV 5%); cGVHD moderate/severe: 15% (MAC) aGVHD II–IV 37.6% (III–IV 10.1%); cGVHD severe 2y: 27% (MAC) PTCy with peripheral blood stem cells source aGVHD II–IV: 53.8% (III–IV: 6.3%); cGVHD 1y: 21.9% vs. aGVHD II–IV: 51.9% (III–IV: 14.7%); cGVHD 1y: 35.1% PTCy with bone marrow or blood stem cells source aGVHD II–IV: 41% (III–IV: 17%); cGVHD mod/sev 2y: 31% vs. aGVHD II–IV: 48% (III–IV: 18%); cGVHD mod/sev 2y: 47% vs. aGVHD II–IV: 28% (III–IV: 9%); cGVHD mod/sev 2y: 44% aGVHD II–IV: 29–33% (III–IV: 9–10%); cGVHD 2y: 27–33% vs. aGVHD II–IV: 29–32% (III–IV: 4%); cGVHD 2y: 29–25% aGVHD II–IV: 27.9% (III–IV: 7.7%); cGVHD extensive 2y: 14% vs. aGVHD II–IV: 38.3% (III–IV: 15.9%); cGVHD extensive 2y: 14% aGVHD II–IV: 33% (III–IV: 0%); cGVHD 1y: 21; 2y: 23% vs. aGVHD II–IV: 40% (III–IV: 5%); cGVHD 1y: 14; 2y: 19% HR aGVHD II–IV 1.01 (III–IV: 1.38); cGVHD: 1.05; extensive 1.11 HR aGVHD II–IV 1.67 (III–IV: 1.82); cGVHD: 1.46; extensive 1.44	[76] [79] [26] [80] [54] [52] [57] [51] [26]
Immunomediated in vivo T-cell depletion MRD/MUD MUD_RIC	ATG: polyclonal heterologous antibodies Targets: CD2,CD3,CD4, CD8, CD11a, CD107a, CD28 Alemtuzumab (humanized anti-CD-3 monoclonal antibody)	aGVHD II–IV 18% (range 7–34%) cGVHD 28% (range 16–28%) - 2y cGVHD: 32.2% in ATG-F group vs. 68.7% - aGVHD II–IV: 23% in ATG-F group vs. 40%; cGVHD moderate–severe 12% in ATG-F group vs. 33% - aGVHD II–IV: 13.7% in ATG-Thymo group vs. 27% 2y cGVHD: 28% vs. 52.5% in controls - 2y cGVHD: 26.3% with ATG-Thymo vs. 41.3% - 3y extensive cGVHD 12.2% in ATG-F group vs. 45% Alemtuzumab/CyA vs. TMS: severe cGVHD 1y 0% vs 10.3%, 5y 4.5% vs. 28.5%	[32] [67] [68] [69] [70] [71] [75]
Blocking alloreactivity and augmenting TReg MRD/MAC_Tac/Sir vs. Tac/MTX MRD/MUD_RIC TMS vs. CyA/MMF MUD_NMA Sir/CyA/MMF vs. CyA/MMF	Sirolimus mTOR inhibition	Tac/Sir: GVHD II–IV 26%; cGVHD: 53% vs. Tac/MTX: GVHD II–IV 34%; cGVHD: 45% TMS aGVHD II–IV: 9% (III–IV: 3%); cGVHD 2y 59% CyA/MMF aGVHD II–IV: 25% (III–IV: 4%) cGVHD 2y 63% Sir/CyA/MMF aGVHD II–IV: 26% (III–IV: 2%); cGVHD 1y: 49% vs. CyA/MMF aGVHD II–IV: 52%; (III–IV: 8%); cGVHD 1y: 50%	[58] [59] [60]
Ex-vivo alloreactive T-Cell Depletion Haploidentical	selection of CD34+ cells Immune depletion of alpha/beta T cells	aGVHD III–IV 3%; cGVHD 0% aGVHD II–IV: 18% (range 11–28%); cGVHD: 8% aGVHD I–II: 26%; III–IV: 6%; cGVHD: 6%	[81] [85] [86]
Ex-vivo alloreactive T-Cell Depletion Pediatric patients	selection of CD34+ cells CD34+ Megadose (19–21 × 10^6^/kg) + higher T cell dose (1.4–4.7 × 10^4^/kg)	aGVHD 0%; cGVHD 0% aGVHD II–IV: 17%; cGVHD: 35%	[82] [83]
Selective depletion of alloreactive T cells Matched related (adults) Haploidentical (pediatrics) Developmental	Depletion of CD45RA naïve T cells, preserving the CD34+ and CD45RO fractions (better activity vs. infections and GVL)	aGVHD II–IV 66%; cGVHD 7–9% aGVHD II–IV 18%; cGVHD 35%	[84]
↓ T-Cell Co-stimulatory/ Co-inhibitory signal MUD_ CNI/MTX + ABA vs. CNI/MTX + placebo Mismatched CNI/MTX + ABA vs. control IBMTR	Abatacept (CD28/CTLA-4 inhibitor) blockade of costimulatory T cell signaling and inhibition of T cell activation	aGVHD II–IV: 43.1% (III–IV: 6.8%); cGVHD moderate–severe 1y: 44.8% vs. aGVHD II–IV: 62.1% (III–IV: 14.8%) cGVHD moderate–severe 1y: 36% aGVHD II–IV: 41.9% (III–IV: 2.3%); cGVHD moderate–severe: 57.9% vs. aGVHD II–IV: 53.2% (III–IV: 30.2%)	[89]
Blocking T-Cell trafficking (anti-homing compounds) MUD or Mismatched	Vedolizumab α4β7integrins inhibitor: mediates migration of T-cells to intestinal mucosa	aGVHD II–IV: 19% (III–IV: 5%) Lower GI aGvHD: 7.1% vs. 18.8% in the placebo group	[35] [95]
Blocking T-Cell trafficking MRD/MUD RIC	Maraviroc CCR5 inhibitor blocks alloreactive lymphocyte chemotaxis to intestinal mucosa	3 prophylaxis regimens compared in 273 randomized patients 1-TAC + MMF + PTCy; 2-TAC + MTX + BOR; 3-TAC + MTX + Maraviroc No sign of reduction of both acute and cGVHD with Maraviroc	[93]
Modulation of microbiota *Developmental*	Microbiota Transplantation induces modifications of alloreactivity against intestinal mucosa antigens	aGVHD III/IV: 0% vs. 25% in the control group	[96]

MUD: matched unrelated donor; NRM: non-relapse mortality; MRD: matched related donor; CNI, calcineurin inhibitor; MMF: mycophenolate mofetil; CyA: cyclosporine A; MTX: Methotrexate; TCR, T cell receptor; Tac: Tacrolimus; HSCT, hemopoietic stem cell transplantation; NMA, non-myeloablative conditioning; MAC, myeloablative conditioning; PTCy: Post-Transplant Cyclophosphamide; RIC: reduced intensity conditioning; NMA: non myeloablative condiotioning; BM: bone marrow; PBSC: peripheral blood stem cell; ATG: anti-lymphocyte globulin; Thymo, Thymoglobuline; ATG-F: ATG Fresenius; TMS: Tacrolimus/Methotrexate/Sirolimus; Sir: Sirolimus; T reg, regulatory T cells; GVL: graft versus leukemia; ABA: Abatacept; IBMTR: International Bone Marrow Transplant Registry; GI aGVHD: gastrointestinal aGVHD; BOR: Bortezomib; * 2 doses PTCy; ** 1 dose PTCy.

**Table 4 cells-13-01524-t004:** Grading of aGVHD based on the main proposals starting from the first classification by Glucksberg up to the recent one developed by the MAGIC consortium.

Overall Glucksberg Criteria/MAGIC criteria	Original Glucksberg Criteria	Modified Glucksberg or Keystone Criteria	Minnesota Criteria	MAGIC Criteria	IBMTR Criteria	Overall IBMTR Grade
0	no organ involvement (skin = 0; and liver = 0; and GI = 0) corresponds to the absence of aGVHD					O
I	Skin = stage 1–2, without liver/GI involvement or decrease in performance status/fever	skin = stage 1–2, without liver/GI involvement	Skin = stage 1–2, without liver/GI involvement	Skin = stage 1–2, without liver/GI involvement	Skin = stage 1–2, without liver/GI involvement	A
II	Skin = stage 1–2 and (liver and/or GI involvement = stage 1–2) with a mild decrease in performance status	Skin = stage 3 and/or liver = 1 and/or GI = 1	Skin = stage 3 and/or liver = stage 1 and/or GI = stage 1	Skin = stage 3 and/or liver = stage 1 and/or GI = stage	Skin = stage 2 and/or liver = stage 1–2 and/or GI = stage 1–2	B
III	(skin and/or liver and/or GI = stage 2–4) with marked decrease in performance status	Liver = stage 2–3 and/or GI = 2–4	Liver = stage 2–4 and/or GI = 2–3	Liver = stage 2–3 and/or lower GI = 2–3, with skin stage 0–3 and/or upper GI stage 0–1	Skin = stage 3 and/or liver = stage 3 and/or GI = stage 3	C
IV	(skin and/or liver and/or GI = stage 2–4) with Karnofsky < 30%	Skin = stage 4 and/or liver = stage 4	Skin = stage 4 and/or GI = stage 4	Skin = stage 4; and/or liver = stage 4; and/or lower GI = stage 4, with stage 0–1 upper GI	Skin = stage 4; and/or liver = stage 4; and/or GI = stage 4	D

GI, gastro-intestinal tract; aGVHD, acute graft versus host disease; IBMTR, International Bone Marrow Transplantation Registry; MAGIC, Mount Sinai Acute International Consortium.

**Table 5 cells-13-01524-t005:** Severity score of cGVHD based on the main proposals starting from the first classification by Shulman up to the recent one developed by NIH Consensus.

Original Seattle Classification [144]	NIH 2005 [3]	NIH 2015 [4]
**Limited** One or both of: Localized skin involvement Hepatic dysfunction due to cGVHD **Extensive** One of: Generalized skin involvement Localized skin involvement and/or hepatic dysfunction due to GVHD, AND: Liver histology showing chronic aggressive hepatitis, bridging necrosis, or cirrhosis, or: Involvement of eye (Schirmer’s test with <5 mm wetting), OR: Involvement of minor salivary glands or oral mucosa demonstrated on labial biopsy, OR: Involvement of any other target organ	**Mild cGVHD:** only 1 or 2 organs (except the lung), with a score of 1 **Moderate cGVHD:** - At least 1 organ with a score maximum of 2 - Or 3 or more organs with a score of 1 - Only lung score 1 **Severe cGVHD:** - Any organ with a score of 3 - Lung score 2 or greater	**Mild cGVHD:** 1 or 2 organs with a maximum score of 1 AND Lung score of 0 **Moderate cGVHD** - 3 or more organs with a maximum score of 1 - Or at least 1 organ (not lung) with a score of 2 - Lung score 1 **Severe cGVHD:** - At least 1 organ with a score of 3 - Lung score of 2 or 3

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
