# Peer review of "Current Approaches for the Prevention and Treatment of Acute and Chronic GVHD"

_cells, 2024, doi:10.3390/cells13181524_

Round 1

Reviewer 1 Report

Comments and Suggestions for Authors

It was a pleasure to read this very informative review. Nevertheless I would have some suggestions to further improve the clinical value.

-          P3, line 78: "DQB1 DRB3/4/5 and others may increase aGvHD" include Citation (14,18,19?).

-          P2, Table 1 and line 65: PTCy and Source – please include more details and literature (some details are found later in the text, refer on this perhaps).

-          P4, line 134: Campath is not used in UK only, it is recommended for instance in ESID guidelines for non-malignant diseases in children (Citation in BMT).

-   P3 line 123: Rabbit ATG preserves reg T cells: has this one study been verified/ reproduced?

-          P5, line 208: please specify "standard prophylaxis" used in this study

-          Table 2, PTCy: the results are somehow conflicting to data in the text; please specify (Literature should be attached to the results line by line)

-          Table 2, Tac/MTX: please comment on the difference in results of the same strategies (different studies, literature?)

-          P8; line 263: please specify “high dose systemic corticosteroids

-          P8, chapter 2, Line 248ff: Cartoon (Figure) about pathophysiology of GvHD may be helpful (therapeutic substances included)

-          Page 8 and general: please include the grading of aGvHD and cGvHD as well as a short comment to problems related to the commonly used strategies of grading

-          Figure 1: This algorithm is very useful. I would suggest to include algorithms for treatment of aGvHD and cGvHD in general (and perhaps BOS); 1st, 2nd, 3rd line, experimental approaches…)

-          Chapter 6: Please add some remarks about local treatment of cGvHD lesions, in particular skin, eyes and genital tract. Please include also a recommendation to diagnose and treat genital and in particular vaginal lesions of cGvHD

-          P11, lines 386 ff: Again (in addition to comment at page 8), a table or cartoon would be helpful to illustrate the "old and new" definitions of a/c GVHD and overlap chronic GvHD

-          Page 12, line 467 ff. Chapter 7 should reorganized, starting with established therapies - ECP, then Ruxo in one chapter, Ibrutinib and Belumosudil (proven by the FDA), followed by other/experimental approaches.

Author Response

First of all we want to thank the two reviewers for the notes and precious suggestions, which we have tried (to the extent possible) to accommodate; below we report our point-by-point detailed answers:

  • P3, line 78: "DQB1 DRB3/4/5 and others may increase aGvHD" include Citation (14,18,19?).

The exact correspondence of the references with the text has been checked 

  • P2, Table 1 and line 65: PTCy and Source – please include more details and literature (some details are found later in the text, refer on this

Done: references about PTCy and BM vs PBSC have been updated  and the table1 has been modified.

  • P4, line 134: Campath is not used in UK only, it is recommended for instance in ESID guidelines for non-malignant diseases in children (Citation in BMT).

We have added to the text this important information and included the suggested reference.

  • P3 line 123: Rabbit ATG preserves reg T cells: has this one study been verified/ reproduced

Yes, it is and we have added 3 more references about this specific aspect.

  • P5, line 208: please specify "standard prophylaxis" used in this study

Done

  • Table 2, PTCy: the results are somehow conflicting to data in the text; please specify (Literature should be attached to the results line by line).

Exact matches with references have been verified and corrected in Table 2

  • Table 2, Tac/MTX: please comment on the difference in results of the same strategies (different studies, literature?).

A short comment has been inserted at the end of the paragraph that illustrated the results of the studies about this topic

  • P8; line 263: please specify “high dose systemic corticosteroids”

Done

  • P8, chapter 2, Line 248ff: Cartoon (Figure) about pathophysiology of GvHD may be helpful (therapeutic substances included).

 Done (we have added one figure for aGVHD and another for cGVHD pathogenesis, including the main agents with the main corresponding targetable pathogenetic pathway).

  • Page 8 and general: please include the grading of aGvHD and cGvHD as well as a short comment to problems related to the commonly used strategies of grading.

Done: for both aGVHD and for cGVHD the therapy chapters now have an initial part that illustrates the problems of grading and scoring

  • Figure 1: This algorithm is very useful. I would suggest to include algorithms for treatment of aGvHD and cGvHD in general (and perhaps BOS); 1st, 2nd, 3rd line, experimental approaches…).

We  tried to do our best by adding two original algorithms for BOS and for experimental approaches in this field (see fig 4).

  • Chapter 6: Please add some remarks about local treatment of cGvHD lesions, in particular skin, eyes and genital tract. Please include also a recommendation to diagnose and treat genital and in particular vaginal lesions of cGvHD.  

We have added a brief paragraph including the clinical manifestations and the ancillary treatments for these particular anatomical districts.

  • P11, lines 386 ff: Again (in addition to comment at page 8), a table or cartoon would be helpful to illustrate the "old and new" definitions of a/c GVHD and overlap chronic GvHD.

We added a table illustrating the different forms and the old and new definitions of acute, overlap and cGVHD.

  • Page 12, line 467 ff. Chapter 7 should reorganized, starting with established therapies - ECP, then Ruxo in one chapter, Ibrutinib and Belumosudil (proven by the FDA), followed by other/experimental approaches.

Done: the chapter of therapy of cGVHD has been reorganized according to these suggestions.

Reviewer 2 Report

Comments and Suggestions for Authors

This article mainly reviews the current prevention and treatment methods of acute GVHD and chronic GVHD. I have some comments.

1. This article does not cover the latest research progress. For example, MSC cell therapy has made rapid progress in the prevention and treatment of GVHD, but there are few relevant data descriptions in this article.

2. The logic of this article is poor: risk factors have no logical relationship with the prevention and treatment of GVHD; the pathophysiological mechanism, drug resistance mechanism and prognosis were described in the treatment part, and there was no logical connection with the treatment content, so it was suggested to adjust its position in the article.

3. The introduction part is short, not general, and does not outline the purpose of the article.

4. The author's point of view is unclear, and the conclusion has a poor connection with the full paper.

5. Specific content:

Appropriate segmentation of Risk factors of acute and cGVHD; to determine whether PTCy is a risk factor for GVHD.

The mechanism of Alemtuzumab should be briefly described in the GVHD prevention section.

The Therapy of aGVHD part needs to be supplemented with data related to hepatic GVHD.

In the section on Therapy of SR-cGVHD, it is suggested that the order of several drugs reviewed be adjusted. Ibrutininb should not be reviewed. There have been too few recent advances in second-line therapies. In addition, this article should focus on the overall prevention and treatment of GVHD and not overreview the effect of an organ.

The figure summary should be supplemented in the full paper.

Comments on the Quality of English Language

There are no punctuation marks in many places in the manuscript, and the writing of nouns is not uniform. English writing needed to be improved.

Author Response

First of all we want to thank the two reviewers for the notes and precious suggestions, which we have tried (to the extent possible) to accommodate; below we report our point-by-point detailed answers:

1.          This article does not cover the latest research progress. For example, MSC cell therapy has made rapid progress in the prevention and treatment of GVHD, but there are few relevant data descriptions in this article.

The aim of this review is not to cover all the latest research advances, but to offer to the reader the most relevant information about current practice, medical needs and an update about the treatments recently used in phase 2-3 clinical trials, potentially changing the current practice. There are a large number of experimental therapies ongoing, some of which are far from their possible use and addressing them all would have been dispersive. As for the use of MSCs, this is not at all recent, particularly in the context of aGVHD.  Experimental approaches including very preliminary data in the treatment of acute and chronic GVHD have not included or have not been covered extensively to limit the length of the paper. Moreover as regards the  therapy with MSCs, up to now, unfortunately all randomized trials with MSCs have essentially failed (as reported in the Cochrane systematic review by Fisher SA, Cutler A, Doree C et al , 2019); after this systematic review only one randomized study using a commercial MSC product has been reported in the setting of aGVHD, but unfortunately it  failed the end point; therefore it did not seem useful to us to recall dated literature or draw the reader's attention to the several ongoing protocols with MSC, but of which there data are not available yet. However, according to the reviewer suggestions,  we have now expanded this topic by selecting some recent papers in this area, focusing about different MSC manufacturing protocols aimed to achieve higher efficacy, although these results have not been extensively reproduced, also in the few countries in which the MSC use has been approved. New recent data obtained by the most recent trials with MSC both for prevention and the treatment of aGVHD have been also added in this new version of the paper. The use of the microbiota and new experimental approaches have been also expanded including the possible use of Placenta-derived cells as alternative source of immunomodulating cells instead of the MSC (although clinical data about their efficacy are still lacking) and finally we have expanded the references of this topic, also including a recent review focusing this specific aspect (Novel therapies for graft versus host disease with a focus on cell therapies: Zeiser R. et al. Frontiers in Immunology. 05 October 2023 DOI 10.3389/fimmu.2023.1241068)

2.          The logic of this article is poor:

a- risk factors have no logical relationship with the prevention and treatment of GVHD;

b- the pathophysiological mechanism, drug resistance mechanism and prognosis were described in the treatment part, and there was no logical connection with the treatment content, so it was suggested to adjust its position in the article.

a- With all due respect to the reviewer's opinion, we firmly believe that the evaluation of some risk factors has a role in the GVHD prevention strategy (mainly as regard the donor selection and the stem cell source). Obviously, risk factor management is critical for prevention and not for treatment of GVHD; this aspect has been clarified in the current version. Moreover we have separated this chapter in order to avoid a possible misunderstanding.

b- As regards the apparent absence of a logical connection between pathophysiological mechanisms and treatment, we would like to point that the initial purpose of this review (mainly aimed at the clinician reader) was to favor a reasoned approach to treatment taking into account the targets towards which treatment could be directed, as well as the possible mechanisms of drug resistance and the expected outcomes; therefore we initially included in the same paragraph the most relevant pathophisiological pathways, linked with the most relevant target therapies (eg  Ruxolitinib, Belumosudil, Axatilimab).

This is the reason why treatment has been initially linked to pathophysiology and its pathways; nevertheless, we have modified the structure of the review according to the Reviewer preference.

c- The introduction part is short, not general, and does not outline the purpose of the article.

We have implemented this useful suggestion by expanding the introduction and by outlining the article purpose.

3.          The author's point of view is unclear and the conclusion has a poor connection with the full paper.

We have tried to better explain our point of view, trying to better connect the conclusions (which has been expanded) with the previous part.

4.          Specific content: Appropriate segmentation of Risk factors of acute and cGVHD; to determine whether PTCy is a risk factor for GVHD.

We thank the Reviewer for having reported this possible misunderstanding, generated by some contents of Table 1. The table has been now modified by eliminating some inappropriate definitions and at the same time it has been updated with further references in particular to the comparison between the different Stem Cell source (BM vs PBSC) in the PTCy setting.

5.          The mechanism of Alemtuzumab should be briefly described in the GVHD prevention section.

Done

6.          The Therapy of aGVHD part needs to be supplemented with data related to hepatic GVHD.

To our knowledge it appears difficult to specifically address the results of therapies from studies evaluating only patients with hepatic GVHD since there are no published experiences addressing this specific area; the diagnosis of isolated acute hepatic GVHD is a very rare event (6-8%); therefore the response on the hepatic side can be evaluated only from some case series including patients with multiple organ involvement. Notwithstanding we have now included in the paragraph on MSCs  an appropriate reference with a brief comment concerning a post-hoc analysis of a phase 3 trial with Remestemcel-L, reporting that repeated infusions of MSCs  in patients with liver GVHD  involvement, showed significantly higher durable response rate (29% versus 5%; p= 0.047), despite the MSC use failed to show superiority over the placebo.

7.          In the section on Therapy of SR-cGVHD, it is suggested that the order of several drugs reviewed be adjusted. Ibrutininb should not be reviewed.

We adjusted the order of the drugs for SR-cGVHD according both the Reviewer 1 and to the Reviewer 2 suggestions. As for the Ibrutinib exclusion we aknowledge the preference of the Reviewer 2 (but this seems not to be shared by the Reviewer 1) to exclude Ibrutinb, however this drug has been approved by the FDA for refractory cGVHD and therefore excluding it would not have been possible. However, we have specified that this drug has been tested in a phase 2 study in a small series of patients, focusing the critical aspects relating to the infectious risk.

8.          There have been too few recent advances in second-line therapies. In addition, this article should focus on the overall prevention and treatment of GVHD and not overreview the effect of an organ.

We have chosen (and we hope that this point of view should be quite clear to the reader) to focus attention on the main unmet medical needs of the therapy of GVHD which are mainly represented by the lower GIT in the setting of acute GVHD and by lung involvement in the setting of cGVHD; however, general aspects of prevention and treatment have been highlighted as requested; according to the Reviewer 1 and 2 we have also covered  the main aspects of ancillary and topical treatment in anatomical districts generally neglected or poorly addressed (skin eyes, mouth, gynecological involvement)

9.           The figure summary should be supplemented in the full paper. Done: we have added a file including the summary of tables and of the figures

Round 2

Reviewer 1 Report

Comments and Suggestions for Authors

The review is higly imformative and well written. Figures have been added to illustrate pathophysiology and treatment principles. of acute and chronic Tables have been updated as suggested.

I would suggest to add the following treament strategies for refractory GvHD:

- pTCy + CD34 Boost (Kloehn et al., Int J Mol Sci., 2022)

- Campath (Schub et al., BMT, 2010)

Author Response

Comment 1: I would suggest to add the following treament strategies for refractory GvHD:

  • PTCy + CD34 Boost (Kloehn et al., Int J Mol Sci., 2022)
  • Campath (Schub et al., BMT, 2010)

Response 1: We have included the references suggested

Reviewer 2 Report

Comments and Suggestions for Authors

Accept after English writing improvement.

Comments on the Quality of English Language

Accept after English writing improvement.

Author Response

Comment: Accept after English writing improvement

Response: we have been carefully revised both as regards the correctness of grammar, syntax and style